# Transverse barrier formation by electrical triggering of a metal-to-insulator transition

Pavel Salev [1✉], Lorenzo Fratino [2], Dayne Sasaki [3], Rani Berkoun[2], Javier del Valle[1,4], Yoav Kalcheim[1,5], Yayoi Takamura [3], Marcelo Rozenberg [2] & Ivan K. Schuller[1]

Application of an electric stimulus to a material with a metal-insulator transition can trigger a large resistance change. Resistive switching from an insulating into a metallic phase, which typically occurs by the formation of a conducting filament parallel to the current flow, is a highly active research topic. Using the magneto-optical Kerr imaging, we found that the opposite type of resistive switching, from a metal into an insulator, occurs in a reciprocal characteristic spatial pattern: the formation of an insulating barrier perpendicular to the driving current. This barrier formation leads to an unusual N-type negative differential resistance in the current-voltage characteristics. We further demonstrate that electrically inducing a transverse barrier enables a unique approach to voltage-controlled magnetism. By triggering the metal-to-insulator resistive switching in a magnetic material, local on/off control of ferromagnetism is achieved using a global voltage bias applied to the whole device.

[1] Department of Physics and Center for Advanced Nanoscience, University of California San Diego, La Jolla, CA, USA. [2] Université Paris-Saclay, CNRS Laboratoire de Physique des Solides, 91405, Orsay, France. [3] Department of Materials Science and Engineering, University of California Davis, Davis, CA, USA. [4]Present address: Department of Quantum Matter Physics, University of Geneva, Geneva, Switzerland. [5]Present address: Department of Materials Science and Engineering, Technion-Israel Institute of Technology, Haifa, Israel. ✉email: psalev@ucsd.edu

Materials with unique functional properties can replace large sections of complex circuits greatly improving the scalability and energy efficiency of electronic devices[1–4]. For instance, using materials in which voltage application induces resistive switching makes it possible to mimic both synaptic plasticity[5–9] and diverse neuron spiking behaviors[10–13] in circuits consisting of just a few components. In contrast, tens of conventional CMOS transistors are required to achieve similar functionalities[14,15]. Deep understanding of physical properties and their response to external stimuli becomes critical for designing applications using such advanced electronic materials. Many previous studies explored various aspects of nonvolatile switching based on ionic electromigration, which is promising for memory applications[16].

Recently, there has been a great interest in a different type of resistive switching: volatile switching due to electrical triggering of a metal–insulator transition (MIT), i.e. an intrinsic phase transition that alters the charge transport properties of a material (e.g., Mott or Peierls transition). Such volatile switching is induced by applying and holding an electrical stimulus to an MIT material and the switching automatically resets back into the initial state upon turning off the stimulus (hence the term volatile). MIT-based switching is often accompanied by a large change of electrical resistivity and optical properties making it attractive for applications in rf electronics[17,18], optoelectronics[19–21], and biologically inspired artificial neurons[10–13]. Most commonly, the MIT switching occurs from the initial insulating into a metallic phase (I → M), for example, in $VO_2$, $V_2O_3$ and $V_3O_5$[22–26], $NbO_2$[27,28], and $(Pr,Ca)MnO_3$[29,30]. The general picture of the I → M switching is well established. The application of an electric stimulus causes a local phase transition due to Joule heating and/ or field-induced carrier doping[25,28,31–35]. This local transition often follows a characteristic spatial pattern: the formation of a percolating metallic phase filament serving as a conduit for electric current flow inside the insulating phase matrix[22–24,36]. The filament formation causes strong nonlinearities in the current–voltage (I–V) characteristics such as an S-type negative differential resistance (NDR) when a part of I–V curve has negative $dV/dI$ slope making the overall shape to resemble the letter S. The opposite type of volatile switching, in which an electrical stimulus drives the material from the initial metallic into an insulating (M → I) phase, is a rare phenomenon and an understanding of its microscopic process is lacking. Several works reported that passing a sufficient current can trigger the MIT in select colossal magnetoresistance manganites[37–41]. It remains unknown, however, whether the M → I switching follows any characteristic spatial pattern.

In this work, we show that M → I switching occurs by the formation of an insulating barrier perpendicular to the current flow, in contrast to the metallic filamentary percolation along the current. We observed the barrier formation experimentally and correlated its appearance with the development of an unusual N-type NDR nonlinearity in the I–V characteristics of a device. Using theoretical analysis, we present evidence that this transverse barrier formation is a universal property of M → I switching, making our findings broadly relevant to a whole class of such resistive switching systems. Finally, we discuss the implications of our finding to nonvolatile ionic-migration-based low- to high-resistance switching.

## Results

We study volatile resistive switching based on electrical triggering of an MIT in $La_{0.7}Sr_{0.3}MnO_3$ (LSMO) thin film devices (fabrication details are available in "Methods" section and in Supplementary Information 1). Under equilibrium conditions (i.e. without application of high voltage/current), the devices have two coupled phase transitions at $T_c \approx 340$ K: from a low-temperature ferromagnetic metal to a high-temperature paramagnetic insulator (Fig. 1a). The coupling between the two transitions, magnetic and MIT, is mediated by the double exchange mechanism[42,43]. The fact that the transitions occur simultaneously is the key property that allowed us to map the spatial distribution of resistive switching as discussed later in the paper.

Resistive switching in LSMO manifests as strong nonlinearities in the I–V characteristics. Current-controlled I–V curves (Fig. 1b) show an abrupt and hysteretic switching from a low- to a high-resistance state above a temperature-dependent current threshold. The switching is volatile, i.e. the device automatically resets into the initial low-resistance state when the current is ramped down. Distinct switching, i.e. an abrupt jump in an I–V curve, can be observed at all temperatures below 310 K. As the temperature approaches the phase transition, in 310–340 K range, the I–V curves are nonlinear but do not display an abrupt discontinuity. Above the transition ($T > T_c \approx 340$ K), the switching completely disappears and the I–V curves become linear indicating the close relation between the switching and the MIT: the metal-to-insulator switching cannot be induced when the LSMO device is already in the insulating state. In contrast to the abrupt switching under current biasing, the voltage-controlled I–V curves (Fig. 1c) are smooth and display an unusual N-type NDR (the I–V shape resemble the letter N). The NDR region develops when the device actively undergoes resistive switching: as the applied voltage is ramped up, the current decreases, indicating that the resistance progressively increases. The detailed discussion about the origin of volatile resistive switching in LSMO is available in Supplementary Information 2. We compared the resistance values in I–V curves to resistance–temperature (R–T) dependence, tested the influence of the switching on the R–T dependence, analyzed the switching power vs. temperature and the switching voltage vs.

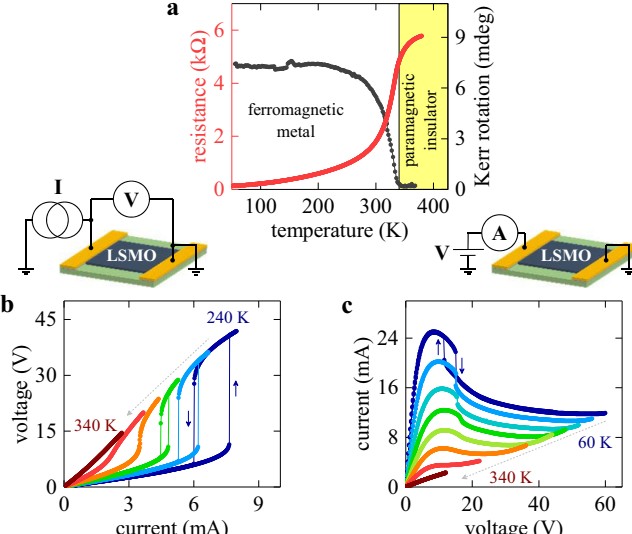

**Fig. 1 Volatile metal-to-insulator resistive switching. a** Metal–insulator (red line) and magnetic (gray line) transitions in a 50 × 100 μm² $La_{0.7}Sr_{0.3}MnO_3$ (LSMO) device probed by electrical transport and MOKE measurements. **b** Current-controlled I–V curves showing an abrupt and hysteretic volatile metal-to-insulator resistive switching. The measurements are in the 240–340 K temperature range with a step of 20 K (color-coded from blue to dark red). **c** Voltage-controlled I–V curves showing a gradual resistive switching and an N-type NDR region. The measurements are in 60–340 K temperature range with a step of 40 K (color-coded from blue to dark red).

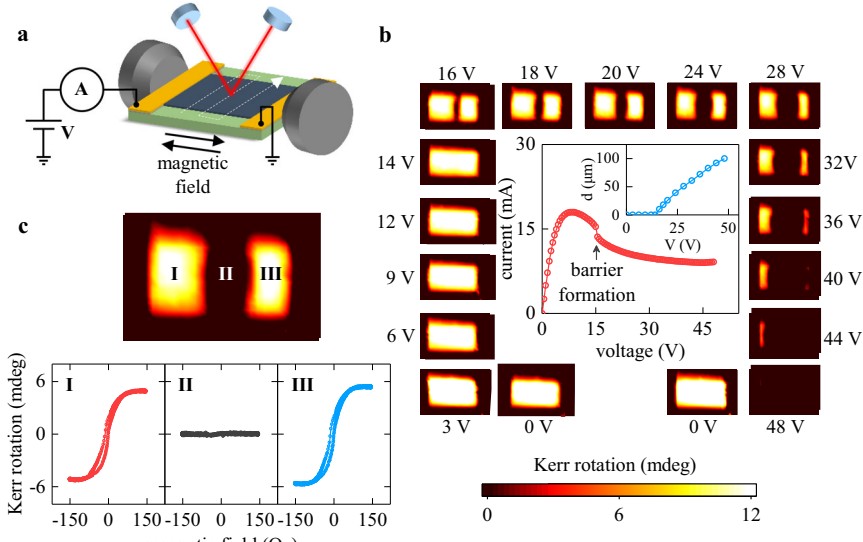

**Fig. 2 Spatial mapping of the metal-to-insulator resistive switching. a** Schematic of the MOKE measurement setup. The MOKE hysteresis loops were acquired at every *xy*-spot in the device area. The magnetic field was applied in-plane along the device length. Voltage biasing was maintained without interruptions over the entire measurement time. **b** Simultaneously recorded *I*–*V* curve (center) and MOKE *xy*-maps (sides). The bright areas in the maps correspond to the ferromagnetic LSMO. The total field of view is 90 × 140 μm². In the maps, the electric current flows horizontally. As the *I*–*V* progresses through the NDR, a transverse insulating paramagnetic barrier appears in the device center and keeps expanding with increasing the applied voltage. The inset in the *I*–*V* plot shows the barrier size, *d*, as a function of applied voltage, *V*. **c** MOKE map and local hysteresis loops corresponding to three device regions (labeled using Roman numerals) recorded at 24 V. While the device sides (regions I and III) show ferromagnetic response, the MOKE signal is zero in the center (region II). All measurements were done at 100 K.

temperature dependencies, tested for bipolar switching, performed cycling and voltage stress tests, and probed the influence of oxygen partial pressure on the switching behavior. All our results point to the single conclusion that the observed volatile switching in LSMO originates from the MIT triggering mediated by Joule heating, which is a common switching mechanism in MIT-based devices[28,44,45]. We found no evidence that ionic migration, which under special conditions could produce volatile threshold switching[46], play any significant role in our experiments. Overall, the volatile MIT-based resistive switching in LSMO provides two distinct operation modes (the abrupt switching using current biasing vs. the gradual switching using voltage biasing) and does not lead to irreversible chemical or structural changes of the material, which could cause highly inhomogeneous electrical property distribution. In addition, LSMO, unlike many other members of the manganite family, does not have intrinsic metal/insulator phase separation below $T_c$[47], which could create preferential current flow channels. Thus, LSMO is an ideal material choice to explore the fundamental properties of the M → I switching process.

To understand the underlying microscopic mechanism of the MIT resistive switching, we performed *in operando* imaging of the LSMO devices exploiting the fact that the MIT occurs simultaneously with the magnetic transition. Using scanning magneto-optical Kerr effect (MOKE) microscopy (Fig. 2a), we mapped the spatial distribution of ferromagnetic regions while applying a voltage bias. The measurement procedure involved recording MOKE hysteresis loops at every spot over the device area using a 5-μm-size laser beam. We represent the data by plotting *xy*-maps of the MOKE loops magnitudes (i.e. the maximum Kerr rotation angle). We note that our MOKE maps are different from conventional MOKE images in which the contrast originates from domains of different magnetization orientation. In our case, the bright areas correspond to ferromagnetic regions, while the dark areas indicate the absence of ferromagnetism.

We found that M → I resistive switching occurs by the formation of an insulating barrier that spans across the entire device width in the direction perpendicular to the electric current flow. Figure 2b shows the MOKE maps at different voltages and the corresponding *I*–*V* curve. The device remains uniformly ferromagnetic (metallic) below 15 V, but applying a higher voltage causes the LSMO to transform into a qualitatively different state. At 16 V, the *I*–*V* curve displays a small jump and simultaneously a ~5-μm-wide nonmagnetic domain appears near the device center. The domain spans laterally across the full device width and its size increases with applied voltage until it encompasses the entire device at 48 V (inset in the *I*–*V* plot in Fig. 2b). Because of the direct correspondence between the magnetic and electric properties in LSMO, our measurements imply that the resistive switching from a metal into an insulator does not occur uniformly throughout the device. Instead, an unusual out-of-equilibrium phase separation is favored: a transverse insulating barrier divides the conducting matrix and blocks the electric current flow. This is reciprocal to the resistive switching from an insulator into a metal (for example, in vanadium oxides[22–24,48]), which occurs by the formation of longitudinal percolating metallic filaments. The appearance of a voltage-induced insulating barrier proves that the LSMO device undergoes an abrupt resistive switching on a microscopic level, from the metallic into the insulating phase, even though the global *I*–*V* curve of the entire device displays smooth, gradual, low- to high-resistance evolution. The barrier formation is highly reproducible: multiple devices of different geometries patterned on the same LSMO film and on another film of different thickness showed the same behavior (Supplementary Information 3). On the other hand, we did not find any indications of a barrier appearing during the equilibrium thermal transition (without applied voltage), which proves that the barrier is not simply due to inhomogeneities in the LSMO film (Supplementary Information 4). Therefore, the formation of an insulating barrier is not an accidental device property but a special feature of voltage-driven MIT.

Resistive switching in LSMO enables a unique approach to voltage-controlled magnetism. Voltage-driven metal/insulator/metal phase separation results in an unusual ferromagnetic/paramagnetic/ferromagnetic domain configuration. Figure 2c shows a MOKE map and three local hysteresis loops recorded in the LSMO device under the application of 24 V. While both left and right sides of the device (labeled I and III) have normal ferromagnetic hysteresis loops, the center part (labeled II) does not show any magnetic response: the recorded MOKE loop is just a flat line. The device can serve as an on/off magnetic switch where a local switching is achieved using a global voltage stimulus. For the purpose of imaging, the devices have relatively large dimensions ($50 \times 100\ \mu m^2$) and the minimum observed paramagnetic insulating barrier size was ~5 μm (which also could be convoluted by the 5 μm laser beam size). Our analytical calculations suggest that the minimum stable barrier size could be reduced substantially by reducing the device dimensions and by selecting a material with a high insulator-to-metal resistivity ratio (Supplementary Information 5), potentially allowing local control of magnetic state at nanoscale. As discussed later in the paper, we expect the development of a large thermal gradient associated with the barrier formation. Thus, resistive switching in LSMO could become a useful platform for spin caloritronics[49]. Because of the fundamental nature of the $M \rightarrow I$ switching, the temperature gradient emerges naturally in simple, planar-geometry LSMO devices under a voltage bias, which eliminates the need for external heaters. This simplifies the control over the temperature profile and at the same time provides an easy access for a variety of surface-sensitive techniques such as magnetic force microscopy, magnetic photoelectron emission microscopy, and nitrogen-vacancy centers in diamond.

The insulating barrier formation and the appearance of N-type NDR during the voltage-controlled switching are strongly linked to each other. Neither barrier nor NDR can be observed when the LSMO device is switched using a current bias (Fig. 3). Under current-controlled conditions, the I–V curve displays an abrupt jump and, at the same time, the magnetic signal in the MOKE

maps completely vanishes throughout the entire device. The different behaviors in current- and voltage-controlled switching (full device transition vs. barrier formation) can be understood by considering the impact of the barrier formation on the current and voltage distributions within the device. The formation of a transverse barrier would not disrupt a homogeneous current flow because the barrier spans across the full device width. Therefore, the entire device is able to switch at a threshold current under current-controlled conditions. The situation differs radically under voltage biasing. The formation of an insulating barrier concentrates the applied voltage, which leads to a highly inhomogeneous voltage distribution inside the device, and at the same time impedes current flow, which causes the N-type NDR. The development of this voltage inhomogeneity explains why the current- and voltage-controlled I–V curves differ (abrupt switching vs. N-type NDR) when the device actively undergoes the resistive switching, but the two curves perfectly coincide when the device is either in a uniform metallic or insulating states (I–V plot in Fig. 3).

We achieved an excellent agreement between the experimental results and resistive network simulations (Fig. 4a, b) by considering a realistic resistance vs. temperature dependence and thermal effects due to Joule heating. The details of the simulations are available in Supplementary Information 6. Figure 4c shows local temperatures vs. voltage plots at different positions within

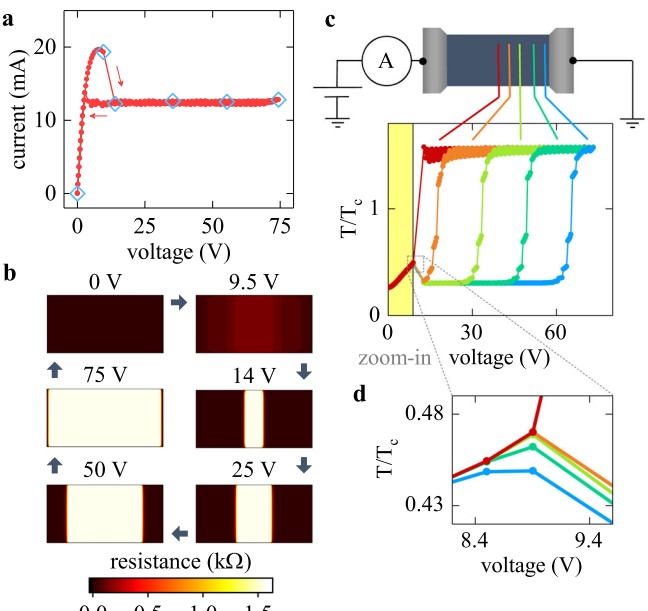

**Fig. 4 Computational analysis of metal-to-insulator resistive switching. a, b** Calculated voltage-controlled I–V curve (**a**) and resistance xy-maps (**b**) showing a remarkable agreement with the experimental data. Diamond symbols in **a** highlight the I–V points for which the resistance maps are shown in **b**. **c** Temperature normalized by the transition temperature, $T/T_c$, vs. applied voltage at several positions along the device length color-coded from red at the center to blue near the electrode. The yellow-shaded region corresponds to the state before the barrier formation. When the barrier appears, the temperature at the device center abruptly increases above $T_c$, while the rest of the device cools down. As the barrier expands with increasing voltage and reaches each point, the temperature at that point increases rapidly above $T_c$. **d** A zoom-in of the temperature vs. voltage plot at the barrier formation. At this magnification, it is clearly visible that just before the barrier appears, the temperature at the center is higher compared to the edges. This thermal inhomogeneity triggers locally the power-temperature positive feedback loop culminating in the insulating barrier formation.

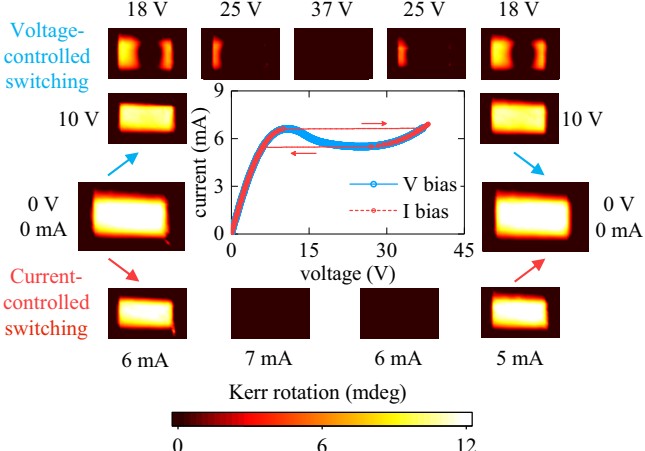

**Fig. 3 Current- vs. voltage-controlled metal-to-insulator switching.** The plot in the center shows two overlaid I–V curves recorded under current- (red) and voltage-controlled (blue) conditions. The I–V curves perfectly coincide outside the NDR region. Corresponding MOKE maps (surrounding the I–V plot) show the switching by the formation and subsequent expansion of a transverse insulating barrier in the voltage-controlled regime (top maps) and the switching throughout the entire device at once in the current-controlled regime (bottom maps). In the maps, the current flows horizontally. The field of view is $90 \times 140\ \mu m^2$. The measurements were done at 250 K.

the device. We found that the applied voltage initially heats up the entire device. This heating, however, is not homogeneous. Just prior to the formation of the insulating barrier, the temperature at the device center is several Kelvins higher compared to the edges (Fig. 4d). This is due to the thermal coupling of the edges to the device electrodes that are at the substrate temperature. Even a small temperature deviation at the center is enough to initiate locally a resistance-power positive feedback loop because of the proximity to the MIT. Higher local temperature increases the resistance, which leads to an increase of the local voltage drop and power dissipation, further increasing the local temperature. As a result, the temperature at the device center abruptly increases well above the $T_c$ and an insulating barrier forms. The barrier concentrates most of the dissipated power, which keeps its temperature high, while the rest of the device cools down almost to the substrate temperature. A distinct and a rather sharp boundary between the two phases can be always observed: the temperature of the metallic regions remains close to the substrate temperature up until those regions are taken over by the expanding barrier as the applied voltage increases. Our calculations predict that the insulating barrier forms exactly at the device center because the extra thermal coupling between the device edges and the electrodes producing a subtle temperature gradient (Fig. 4d) is the only symmetry breaking factor in the model. Experimentally, we observed an asymmetry in the barrier formation position (Figs. 2b and 3, Supplementary Fig. 3 in Supplementary Information 3). Because the barrier formation position does not change upon repeating the switching measurements multiple times, the experimental asymmetry is not stochastic but most likely is due to defects. A region with a locally increased resistivity, due to a subtle temperature gradient or due to defects, would become the hotspot for the barrier formation, but the barrier formation in itself is the direct consequence of the M → I switching. In our model, the only special ingredient enabling the barrier formation is a thermal transition from a low- to a high-resistance state. Because in the simulations we did not have to make any explicit assumption about the phase separation, magnetic properties, or defect density profiles, we conclude that our analysis provides a universal description of the voltage-triggered metal-to-insulator phase transition mediated by Joule heating. Many other materials in the manganite family and some magnetic semiconductors could have similar resistive switching behavior.

## Discussion

From a practical point of view, volatile switching in LSMO has a rare combination of functional properties. Our devices show the switching ratio of $\Delta R/R \sim 300\%$ at room temperature. While higher ratios can be achieved in volatile threshold switches, for example in CBRAM-based devices[50], the switching in LSMO occurs from a low- to a high-resistance state. The low-to-high-resistance switching is a unique feature among the volatile switching devices and it could facilitate the implementation of special functionalities in practical applications. The switching in our devices shows no noticeable cycle-to-cycle variability (Figs. S2.5 and S2.7), which is a common problem in resistive switching devices[51], and has an excellent endurance as no sign of device degradation was observed over an 8-hour-long dc voltage stress test (Supplementary Fig. 2.6) and over $5\times10^6$ fast switching cycles (Supplementary Fig. 2.7). Moreover, resistive switching in LSMO drives the switching of ferromagnetism, which constitutes a special functionality that could be relevant to bridging the gap between conventional charge-based electronics and spintronics. Because of the large device sizes fabricated for MOKE imaging ($50 \times 100$ $\mu m^2$), the switching in our devices requires large driving voltages/currents resulting in a relatively slow switching

speed and large dissipated power, especially at cryogenic temperatures. Reducing the device size down to nanoscale dimensions has been shown to enable sub-nanosecond switching times and dramatically reduce the switching energy down to a few picojoules in MIT-based volatile resistive switching mediated by Joule heating[32,35]. Because MIT-based switching relies on the intrinsic material properties, i.e. triggering of a phase transition, such switching is possible in devices of various geometries including a vertical crossbar[52], a geometry most suitable for dense circuit integration. Nanoscale patterning[53] and fabrication of vertical magnetoresistance devices[54] has been demonstrated using LSMO. Thus, successful implementation of nanoscale vertical resistive switching LSMO devices should be possible as well.

Volatile and nonvolatile resistive switching types bare certain similarities. The formation of percolating conducting filaments occurs both in I → M volatile switching and in nonvolatile high-to-low-resistance switching. The latter can be due to oxygen electromigration in binary[55–63] and complex oxides[64,65] or due to metal cation diffusion in conducting bridge memories based on oxide[66–74] and non-oxide materials[75,76]. Our resistor network simulations show that the insulating barrier that forms during the M → I volatile switching concentrates not only the dissipated power but also the applied voltage. It is possible that the transverse barrier formation can occur in systems which feature a nonvolatile low-to-high-resistance switching caused by ion electromigration, such as rare-earth nickelates[77,78], manganites[77–82], and cuprates[77,79,83]. In fact, spatial patterns previously reported for the nonvolatile switching in $YBa_2Cu_3O_7$[83], LSMO[84] and in $NdNiO_3$[85] can be rationalized by considering the tendency of low- to high-resistance switching to occur by the formation of a transverse insulating barrier.

The results presented in this paper complete the picture of the volatile resistive switching in MIT materials. Two switching types are possible: insulator-to-metal (I→M), e.g. in $VO_2$ or $NbO_2$, and metal-to-insulator (M→I), e.g. in LSMO as shown in this work. These two switching types have contrasting behaviors. Under voltage biasing, I→M is abrupt and hysteretic while M→I is gradual and has an N-type NDR. Under current biasing, I→M occurs gradually with an S-type NDR and M→I is abrupt and hysteretic. In both cases, the gradual switching is accompanied by a spatially inhomogeneous transition occurring in a characteristic pattern. In I→M, this characteristic pattern is a conducting filament parallel to the current flow, while in M→I an insulating barrier perpendicular to the current flow forms during the switching. Combining the two types of resistive switching provides a broad range of nonlinear electrical properties, which greatly enriches the design space for complex-behavior electronic devices. The coupling between the MIT and ferromagnetic transition, which is a special property of LSMO mediated by double exchange mechanism, allows local on/off switching of magnetism by the resistive switching driven transverse barrier formation. The thermal behavior associated with the barrier formation has direct relevance to devices based on spin caloritronics effects.

## Methods

**Sample preparation**. $La_{0.7}Sr_{0.3}MnO_3$ films of 20 nm and 50 nm thickness were grown epitaxially on a (001)-oriented $SrTiO_3$ substrates using pulsed laser deposition with the laser fluence of 0.7 J/cm$^2$ and frequency of 1 Hz. During the growth, the substrate temperature was held at 700 °C with the oxygen pressure of 0.3 Torr. After deposition, the films were slowly cooled to room temperature in 300 Torr $O_2$ to ensure proper oxygen stoichiometry. The electrodes of (100 nm Au)/(20 nm Pd) for electrical measurements were made using standard photolithography process and e-beam evaporation. The bottom Pd layer was used to achieve a low contact resistance with the LSMO film. After the electrode fabrication, 50×100 $\mu m^2$ devices were defined using reactive ion etching in an Ar/Cl$_2$ atmosphere.

**Electrical measurements.** Resistive switching measurements were performed in a Quantum Design PPMS DynaCool cryostat using a Keithley 2450 source meter in either voltage- or current-controlled mode.

**MOKE measurements.** Magneto-optical imaging of resistive switching was performed in a Montana Instruments NanoMOKE 3 system. The light source was a 660 nm laser focused to a 5-μm-size spot. The magnetic field was cycled in the ±150 Oe range at the 4.7 Hz repetition rate. Voltage/current was applied to the LSMO devices using a Keithley 2450 source meter. The measurement procedure was the following. First, a voltage/current was set. Then the laser was focused at a starting $xy$-coordinate and a MOKE hysteresis loop averaged over 20 cycles was recorded. The loop recording was continued at every $xy$-coordinate until the entire imaging area was covered (typically 90×140 μm$^2$). After this, a new voltage/current was set and the loop recording procedure was repeated. We note that the voltage/current was maintained without interruptions during the entire imaging procedure.

## Data availability

The experimental data generated in this study have been deposited in Zenodo database under accession code https://doi.org/10.5281/zenodo.5165080.

## Code availability

The simulation code has been deposited in Zenodo database under accession code https://doi.org/10.5281/zenodo.5165080.

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

## Acknowledgements

This work was supported as part of Quantum Materials for Energy Efficient Neuro-morphic Computing (Q-MEEN-C), an Energy Frontier Research Center funded by the U.S. Department of Energy, Office of Science, Basic Energy Sciences under Award #DE-SC0019273 (fabrication, measurements, simulations). M.R. acknowledges partial support from the French Agence Nationale de la Recherche "MoMA" project ANR-19-CE30-0020 (computational model development).

## Author contributions

P.S. and I.K.S. conceived the experiment, D.S. and Y.T. synthesized the LSMO films, P.S. performed the transport and MOKE measurements, P.S., J.d.V., and Y.K. analyzed the experimental data, L.F., R.B., and M.R. performed the numerical simulations, all authors discussed the results and contributed to the preparation of the manuscript.

## Competing interests

The authors declare no competing interests.
