## [Peer Review File · Nature Communications]

Reviewers' Comments:

Reviewer #1:

Remarks to the Author:

The present manuscript by P. Salev et al reports on an investigation of the sources of Resistive Switching (RS) in the ferromagnetic metallic (FM-M) $\text{La}_{0.7}\text{Sr}_{0.3}\text{MnO}_3$ (LSMO) thin films grown on top of SrTiO_3 (STO) single crystals. Precisely, it deals on Volatile RS (VRS), instead of the most common Non Volatile RS (NVRS), more widely investigated up to now.

This is an issue that has been widely investigated previously by several authors and the present report includes an imaging method of the FM behavior (Magneto-optic Kerr effect) which is, indeed, a novel contribution to the field and thus a thorough consideration of the preexisting information about the RS of this type of FM-M oxides, together with the new results, should contribute to further understanding of this complex problem.

In my opinion, even if I consider that the manuscript provides a significant novel perspective to advance in this field, it lacks completeness and in-depth analysis of previous works in relationship with their own results. I consider that additional experimental characterization should be carried out to reach solid conclusions about the main claim of the manuscript, i.e. that there's a self-heating process in the LSMO films induced by injected currents. This effect generates the nucleation of an insulating region in a specific place (central part between electrical contacts) and which then extends towards the whole film when the voltage is increased. I include below a list of criticisms that the authors should properly consider before any decision is taken about the suitability of the manuscript for publication in Nature Communications.

- The title proposed for the manuscript is not adequate. It doesn't capture precisely the content of the manuscript. The article deals with VRS and this is essentially different from NVRS and so it should be precised. NVRS in LSMO has been widely investigated before and detailed microscopic mechanisms have been described. For the sake of completeness, and to avoid any misunderstanding, I suggest that the studies refereed to NVRS in LSMO and similar oxides displaying Metal – Insulator Transition (MIT) are properly mentioned.
- The term "in reverse" in the title is not fully understood. It would be easier to refer to VRS in oxides displaying MIT. There have been enough cases of materials initially metallic which become insulating when RS occurs (LSMO, $\text{YBa}_2\text{Cu}_3\text{O}_7$, RENiO_3).
- The formation of a transverse insulating layer has been already demonstrated in NVRS of LSMO thin films and it's not particularly worth to stress this in the title. Just to mention a few examples of non-thermal RS reports in oxides exhibiting MIT: Nano Letters 10, 3828 (2010); J. Electroceram 39, 185 (2017); Adv Electron Mater 5, 1800629 (2019); Small 16, 2001307 (2020); ACS Appl Mater. Interfaces 10, 30552 (2018); Microelectronic Engineering 147 (2015) 37–40; ACS Nano 14, 11765 (2020); Thin Solid Films 553 (2014) 7–12.
- Oxygen migration induced by electric fields have been shown to induce MIT in oxides and associated NVRS behavior. This possible origin of RS should be discarded if a new mechanism is being proposed.
- The present manuscript should demonstrate that no bipolar RS exists in the used films, or alternatively, that the VRS observed is indeed new when high voltages are applied but at lower voltages NVRS also exists. Full $V(I)$ curves should be reported including the reverse polarities to see if they are symmetric or not.
- In the case of NVRS in MIT films it has been shown that there exists a dependence of RS on PO_2 , carrier concentration (changing composition, for instance) and Temperature. All these dependences justify the idea that oxygen migration and oxygen vacancies play a key role. It has also been shown that the Mn valence and the local structure is modified (Raman, electrical resistivity with multiple contacts, C-AFM, STM). Also computer simulation of oxygen diffusion effects confirmed this interpretation. The authors should provide experimental evidence that these parameters do not play any role in their experiments.
- The absence of noticeable bistable resistivity (RS) by self-heating in LSMO was previously demonstrated with current densities up to $J = 10^5 \text{ A/cm}^2$, (Phys Rev B 80, 094412 (2009)). The authors should refer to this previous work and demonstrate that they have gone beyond this current density and self-heating degree. Typical estimated heating values were in the range of 10 – 15 K which should generate only slight changes in the local resistivity.

Questions to be addressed about the presented results:

- The insulating regions of Figs. 2 and 3 appear asymmetric. Can the authors explain why?. Is this a systematic result appearing in all the measured samples?. Is it a reproducible phenomenon?

- Modelling of the thermal behavior (S.I.) of the measured thin films should include as well the electrical contacts and the metals used. The temperature gradients can be deeply modified by them (well-known effect in superconducting materials). Actually, the observed asymmetry could be associated to unequal electrical contacts. The asymmetry seems to be the same when increasing and decreasing the current injection. The results of Figure 4 could also be associated to the influence of the electrical contacts.
 - Have the authors tested different substrates, different substrate thickness or different film thickness to use different current densities?. All these experiments could provide useful insight on the validity of the reported thermal model.
 - Another issue which might help to clarify the validity of the thermal model is to assess if there exists dynamic effects, i.e. have the authors seen any dependence on the measured resistance when the current ramps are modified?. Thermal diffusivity towards the substrate and through the current contacts could be modified and so the measured total resistance. I guess that these dynamic measurements are not compatible with MO Kerr effect mapping of FM regions.
- In conclusion, although I think that the authors report an interesting new set of experimental results and analyses of the RS effect in a very widely investigated material (LSMO), I consider that the claim of generating VRS of thermal origin should be better justified. Additional experimental results should be performed and a thorough consideration of the previous studies on the oxides exhibiting MIT should be included and properly considered in the manuscript.

Reviewer #2:

Remarks to the Author:

The manuscript by Salev et al. reports on resistive switching phenomena in LSMO-based devices. The authors discuss the switching mechanism supported by magneto-optical microscopy. The manuscript is clearly written and structured and the topic is suitable for the journal. However, the presented results are not on the level expected and required for publication in Nature Communications.

General Comments

I see neither some contribution to the fundamental knowledge on resistive switching phenomena, nor on the application side. My general impression is that the authors are not entirely aware on the basic requirements and latest achievement in the field of resistive switching memories.

A. There are no resistive switching characteristics that can verify a reliable device operation. The very small hysteresis in Fig. 1b (only for 3 low temperatures) and in Fig. 1c (only for 60 K) is indicating instable resistive states. At 340 K the relation is linear! The presented I-V and V-I curves are only in the positive range, and only one cycle per temperature is presented. I cannot accept these results as a verification for reliable and stable resistive switching. The currents are high (mA range) and the voltage very high (up to 60 V). Basic resistive switching characteristics are missing: endurance, retention, OFF/ON ratio switching time, evtl. multilevel switching, reliable statistics etc..

B. The fact that the devices are initially low resistive and are further reset to off state is not very unusual. The provided MOKE experiments are clear, but provide no new insight in the RS process. Similar visualization has been provided for e.g. Fe:STO (MRS Proceedings, 1691, Mrss14-1691-bb03-09. doi:10.1557/opl.2014.562). Thus, I cannot acknowledge some new detail on the mechanism revealed.

Some technical comments on the manuscript are provided below.

Comments

1. The authors should be aware that VOx, PCMO and NbO2 are only a very small part of RS materials and systems used. Fundamentally important materials such as STO, TaOx, HfOx, etc. are even not mentioned.

2. On page 3 is written: "...We found that the metal-to-insulator resistive switching occurs via nucleation and growth of an insulating barrier that spans..." Nucleation (phase formation) and growth are two completely different processes. I also do not see any experimental evidences that either nucleation, or growth are rate limiting.

3. The authors are not discussing at all the possibility of oxygen exchange that is supposed to occur at these high voltages and currents. Is the possible loss/enrichment of oxygen also relevant to the resistance change?

4. The I-V plot in Fig. 3 is not presenting any resistive switching. It is an ordinary current-voltage dependence with no indication on resistive switching.

5. Most experiments were performed at low temperatures that are not advantageous for applications.

Reply to Reviewers

Manuscript: Transverse barrier formation by electrical triggering of a metal-to-insulator transition

NCOMMS-20-41785

We thank the Reviewers for their detailed comments. They motivated us to make extensive revisions to clarify our message and to expand the manuscript's scope providing a broader perspective on the physics of resistive switching. Among many changes, we have:

1. Revised the discussion about the differences between volatile MIT-based resistive switching studied in this work and nonvolatile switching due to ionic migration to avoid possible misunderstanding.
2. Added new experimental data that support the Joule heating origin of the volatile resistive switching in (La,Sr)MnO₃ (LSMO) and discards the contribution of oxygen electromigration.
3. Expanded the discussion about the connection between our findings and previous works on ionic electromigration.

We hope that the revised manuscript addresses all concerns of the Reviewers and they would find our manuscript suitable for publication in *Nature Communications*.

Reviewer 1

“The present manuscript by P. Salev et al reports on an investigation of the sources of Resistive Switching (RS) in the ferromagnetic metallic (FM-M) La_{0.7}Sr_{0.3}MnO₃ (LSMO) thin films grown on top of SrTiO₃ (STO) single crystals. Precisely, it deals on Volatile RS (VRS), instead of the most common Non Volatile RS (NVRS), more widely investigated up to now. This is an issue that has been widely investigated previously by several authors and the present report includes an imaging method of the FM behavior (Magneto-optic Kerr effect) which is, indeed, a novel contribution to the field and thus a thorough consideration of the preexisting information about the RS of this type of FM-M oxides, together with the new results, should contribute to further understanding of this complex problem.”

We thank Reviewer for the in-depth analysis of our work and we are glad that Reviewer recognizes the novelty of magnetic spatial mapping of volatile resistive switching based on electrical triggering of metal-to-insulator phase transition. We carefully examined the questions raised by the Reviewer and provided detailed answers in the revised manuscript by including an extended literature review, extended results discussions, and new experimental data.

“In my opinion, even if I consider that the manuscript provides a significant novel perspective to advance in this field, it lacks completeness and in-depth analysis of previous works in relationship with their own results.”

As we understand, the Reviewer refers to the ample literature on nonvolatile resistive switching in

oxides caused by electrically driven ionic migration. Our work focuses on volatile switching via Joule-heating-mediated triggering of an MIT. To avoid confusion that could be caused by “mixing” different switching types within a single discussion we added to the introduction paragraph:

Page 1: *“Many previous studies explored various aspects of nonvolatile switching based on ionic electromigration, which is promising for memory applications [16]. Recently, there has been a great interest in a different type of resistive switching – volatile switching due to electrical triggering of a metal-insulator transition (MIT)”.*

To put our results regarding the volatile MIT switching in a broader perspective and to guide the interested reader to the works describing the spatial patterns that emerge during different types of resistive switching we added a new discussion paragraph:

Page 7: *“Volatile and nonvolatile resistive switching types bare certain similarities. The formation of percolating conducting filaments occurs both in $I \rightarrow M$ volatile switching and in nonvolatile high-to-low resistance switching. The later can be due to oxygen electromigration in binary [47–55] and complex oxides [56,57] or due to metal cation diffusion in conducting bridge memories based on oxide [58–66] and non-oxide materials [67,68]. Our resistor network simulations show that the insulating barrier that forms during the $M \rightarrow I$ volatile switching concentrates not only the dissipated power but also the applied voltage. It is possible that the transverse barrier formation can occur in systems which feature a nonvolatile low-to-high resistance switching caused by ion electromigration, such as rare-earth nickelates [69,70], manganites [69–74], and cuprates [69,71,75]. In fact, spatial patterns previously reported for the nonvolatile switching in $YBa_2Cu_3O_7$ [75], LSMO [76] and in $NdNiO_3$ [77] can be rationalized by considering the tendency of low- to high-resistance switching to occur by the formation of a transverse insulating barrier”.*

“I consider that additional experimental characterization should be carried out to reach solid conclusions about the main claim of the manuscript, i.e. that there’s a self-heating process in the LSMO films induced by injected currents.”

We provided new experimental data in SI 2 decisively showing that the volatile switching in our LSMO devices is due to the triggering of MIT mediated by Joule heating, while oxygen electromigration does not have any significant impact. We direct the reader to SI 2 in the main text:

Page 3: *“The detailed discussion about the origin of volatile resistive switching in LSMO is available in SI 2. We compared the resistance values in I - V curves to resistance-temperature (R - T) dependence, tested the influence of the switching on the R - T dependence, analyzed the switching power vs. temperature and the switching voltage vs. temperature dependencies, tested for bipolar switching, performed cycling and voltage stress tests, and probed the influence of oxygen partial pressure on the switching behavior. All our results point to the single conclusion that the observed volatile switching in LSMO originates from the MIT triggering mediated by Joule heating, while no indication of electrically driven oxygen migration has been found”.*

The details of new experimental data are discussed in the replies to specific questions below.

Specific questions raised by Reviewer 1:

1. “The title proposed for the manuscript is not adequate. It doesn’t capture precisely the content of the manuscript. The article deals with VRS and this is essentially different from NVRS and so it should be precised. NVRS in LSMO has been widely investigated before and detailed microscopic mechanisms have been described. For the sake of completeness, and to avoid any misunderstanding, I suggest that the studies refereed to NVRS in LSMO and similar oxides displaying Metal – Insulator Transition (MIT) are properly mentioned.

The term “in reverse” in the title is not fully understood. It would be easier to refer to VRS in oxides displaying MIT. There have been enough cases of materials initially metallic which become insulating when RS occurs (LSMO, YBa₂Cu₃O₇, RENiO₃).”

We acknowledge that the original title could be confusing. We changed the title to “Transverse barrier formation by electrical triggering of a metal-to-insulator transition”. This new title specifies the resistive switching mechanism (MIT triggering) and explicitly highlights our main conclusion – the formation of a transverse insulating barrier as the characteristic feature of the metal-to-insulator switching.

2. “The formation of a transverse insulating layer has been already demonstrated in NVRS of LSMO thin films and it’s not particularly worth to stress this in the title. Just to mention a few examples of non-thermal RS reports in oxides exhibiting MIT: Nano Letters 10, 3828 (2010); J. Electroceram 39, 185 (2017); Adv Electron Mater 5, 1800629 (2019); Small 16, 2001307 (2020); ACS Appl Mater. Interfaces 10, 30552 (2018); Microelectronic Engineering 147 (2015) 37–40; ACS Nano 14, 11765 (2020); Thin Solid Films 553 (2014) 7–12.”

Understanding the characteristic spatial patterns that emerge in resistive switching is an important topic. The original papers that described the filament formation in various materials received hundreds and even thousands of citations (for example, *Nat. Nanotechnol.* **5**, 148 (2010) - 1936 citations). In this work, we demonstrate that the volatile metal-to-insulator switching occurs in a characteristic pattern, transverse insulating barrier, which is reciprocal to the percolating filament pattern that often occurs in volatile insulator-to-metal switching. Imaging of the volatile metal-to-insulator switching has not been done before. The papers pointed out by the Reviewer describe nonvolatile switching based on ionic electromigration, which is a different switching type. In addition, those papers do not claim the observation of well-defined patterns. We highlight the above points throughout the manuscript:

Page 2: “*It remains unknown, however, whether the $M \rightarrow I$ switching follows any characteristic spatial pattern. In this work, we show that $M \rightarrow I$ switching occurs by the formation of an insulating barrier perpendicular to the current flow, in contrast to the metallic filamentary percolation along the current. We observed the barrier formation experimentally and correlated its appearance with the development of an unusual N-type negative differential resistance (NDR) nonlinearity in the I-V characteristics of a device. Using theoretical analysis, we present evidence that this transverse barrier formation is a universal property of $M \rightarrow I$ switching, making our findings broadly relevant to a whole class of such resistive switching systems. Finally, we discuss the implications of our finding to nonvolatile ionic-migration-*

based low- to high-resistance switching”.

Page 6: “In our model, the only special ingredient enabling the barrier formation is a thermal transition from a low- to a high-resistance state. Because in the simulations we did not have to make any explicit assumption about the phase separation, magnetic properties, defect density profiles, etc., we conclude that our analysis provides a universal description of the voltage-triggered metal-to-insulator phase transition mediated by Joule heating. Many other materials in the manganite family and some magnetic semiconductors could have similar resistive switching behavior”.

Page 7: “Our resistor network simulations show that the insulating barrier that forms during the $M \rightarrow I$ volatile switching concentrates not only the dissipated power but also the applied voltage. It is possible that the transverse barrier formation can occur in systems which feature a nonvolatile low-to-high resistance switching caused by ion electromigration, such as rare-earth nickelates [69,70], manganites [69–74], and cuprates [69,71,75]. In fact, spatial patterns previously reported for the nonvolatile switching in $YBa_2Cu_3O_7$ [75], LSMO [76] and in $NdNiO_3$ [77] can be rationalized by considering the tendency of low- to high-resistance switching to occur by the formation of a transverse insulating barrier”.

3. “Oxygen migration induced by electric fields have been shown to induce MIT in oxides and associated NVRS behavior. This possible origin of RS should be discarded if a new mechanism is being proposed.”

We added new experimental data in SI 2. We found no evidence that oxygen migration occurs in our LSMO devices. All our measurements point to a single conclusion that Joule heating is responsible for the volatile resistive switching in LSMO.

4. “The present manuscript should demonstrate that no bipolar RS exists in the used films, or alternatively, that the VRS observed is indeed new when high voltages are applied but at lower voltages NVRS also exists. Full $V(I)$ curves should be reported including the reverse polarities to see if they are symmetric or not.”

We included full I-V curves in SI 2 (Fig. S2.4). We did not observe bipolar switching neither in the virgin devices nor in the devices that have been cycled many times.

Nonvolatile resistive switching can be induced in our devices by first performing a **high-voltage** electroforming and then cycling the device in moderate voltages. Before the high-voltage electroforming, only volatile MIT-based switching can be observed. The detailed discussion about the possibility of nonvolatile switching is available in SI 2 (Fig. S2.8 and the corresponding text).

5. “In the case of NVRS in MIT films it has been shown that there exists a dependence of RS on PO_2 , carrier concentration (changing composition, for instance) and Temperature. All these dependences justify the idea that oxygen migration and oxygen vacancies play a key role. It has also been shown that the Mn valence and the local structure is modified (Raman, electrical resistivity with multiple contacts, C-AFM, STM). Also computer simulation of oxygen diffusion effects confirmed this interpretation. The authors should provide experimental evidence that these parameters do not play any role in their experiments.”

We tested the volatile resistive switching in our devices in high vacuum and in air. The data is shown in SI 2 Fig. S2.7. We found that the presence of environmental oxygen has no significant impact on the switching behavior.

6. “The absence of noticeable bistable resistivity (R_S) by self-heating in LSMO was previously demonstrated with current densities up to $J = 10^5$ A/cm², (Phys Rev B 80, 094412 (2009)). The authors should refer to this previous work and demonstrate that they have gone beyond this current density and self-heating degree. Typical estimated heating values were in the range of 10 – 15 K which should generate only slight changes in the local resistivity.”

We added to SI 2:

Page 8: “We note that inducing the volatile resistive switching in our LSMO devices required the currents ranging from ~4 mA near room temperature to ~25 mA at 60 K. For a $50 \times 100 \times 0.02 \mu\text{m}^3$ device, those current corresponds to the current density of $0.4\text{-}2.5 \times 10^6$ A/cm². The current densities used in our work are considerably larger compared to the previous work that reported the absence of Joule heating mediated switching in LSMO in $10^4\text{-}10^5$ A/cm² range [10]”.

7. “The insulating regions of Figs. 2 and 3 appear asymmetric. Can the authors explain why?. Is this a systematic result appearing in all the measured samples?. Is it a reproducible phenomenon?”

The insulating barrier typically appears close to the device center, but indeed not exactly at the center. For a specific device, the position where the barrier appears is always the same across multiple switching cycles. In different devices, the barrier position could be different (for example, see Fig. 2 and 3 in the main text and Fig. S3 in SI), but the barrier typically appear some place inside the device, not at the electrodes. This asymmetry in the barrier position is most likely due to defects either in the LSMO film or defects introduced during the fabrication (lift-off and etching). We added to the main text:

Page 6: “Our calculations predict that the insulating barrier forms exactly at the device center because the extra thermal coupling between the device edges and the electrodes producing a subtle temperature gradient (Fig. 4d) is the only symmetry breaking factor in the model. Experimentally, we observed an asymmetry in the barrier formation position (Figs. 2b and 3, Fig. S3 in SI). Because the barrier formation position does not change upon repeating the switching measurements multiple times, the experimental asymmetry is not stochastic but most likely is due to defects. A region with a locally increased resistivity, due to a subtle temperature gradient or due to defects, would become the hotspot for the barrier formation, but the barrier formation in itself is the direct consequence of the $M \rightarrow I$ switching”.

We note that the barrier formation asymmetry is an interesting topic for further research. In fact, we are currently exploring the possibility of pre-defining the barrier formation position and its expansion direction by controllably introducing local defects. The fact that the Reviewer brings up this topic shows us that there is indeed interest in such a work.

8. “Modelling of the thermal behavior (S.I.) of the measured thin films should include as well the electrical contacts and the metals used. The temperature gradients can be deeply modified by

them (well-known effect in superconducting materials). Actually. The observed asymmetry could be associated to unequal electrical contacts. The asymmetry seems to be the same when increasing and decreasing the current injection. The results of Figure 4 could also be associated to the influence of the electrical contacts.”

The effect of electrodes was discussed in the original manuscript and the results in Fig. 4d were indeed attributed to the influence of the electrodes:

Page 6: “Fig. 4c shows local temperatures vs. voltage plots at different positions within the device. We found that the applied voltage initially heats up the entire device. This heating, however, is not homogeneous. Just prior to the formation of the insulating barrier, the temperature at the device center is several Kelvins higher compared to the edges (Fig. 4d). **This is due to the thermal coupling of the edges to the device electrodes that are at the substrate temperature.** Even a small temperature deviation at the center is enough to initiate locally a resistance-power positive feedback loop because of the proximity to the MIT. Higher local temperature increases the resistance, which leads to an increase of the local voltage drop and the power dissipation, further increasing the local temperature. As a result, the temperature at the device center abruptly increases well above the T_c and an insulating barrier forms”.

With regards to electrodes and barrier asymmetry, imperfections at the electrodes potentially could be responsible for the observed asymmetry. Such imperfections fall under the category of defects to which we attribute the asymmetry.

9. “Have the authors tested different substrates, different substrate thickness or different film thickness to use different current densities?. All these experiments could provide useful insight on the validity of the reported thermal model.”

As was discussed in SI 3 of the original manuscript, similar switching behavior was observed in a film of different thickness:

Page 9: “Fig. S3 shows the MOKE maps at different voltages and the corresponding I-V curves recorded in the same sample as in the main text but in a different device (panel a) **and in another LSMO sample of different film thickness (50 nm) and different device dimensions (50×50 μm, panel b).** In both cases, we observed the same behavior as described in the main text: the switching from a metal into an insulator occurs by the formation of an insulating barrier that spans through the entire device width in the direction perpendicular to the current flow. The repeatability of the switching behavior indicates that the formation of an insulating barrier is a general property of the metal-to-insulator switching”.

We also point the reader to SI 3 in the main text:

Page 4: “The barrier formation is highly reproducible: multiple devices of different geometries patterned on the same LSMO film and on another film of different thickness showed the same behavior (SI 3)”.

Testing the effects of different substrates, different crystallographic orientations, different Mn/Sr ratios, etc. may indeed be interesting as these parameters have great impact on both

transport and magnetic properties of LSMO. However, this requires new fabrication and measurement procedures. For example, growing LSMO on a LaAlO_3 substrate induces perpendicular magnetic anisotropy, which requires a different measurement setup than the longitudinal MOKE used in this work. Such studies are beyond the scope of this manuscript. We believe that the extended discussion and new data in SI 2 present enough evidence that Joule heating drives the observed volatile resistive switching in our LSMO devices.

10. “Another issue which might help to clarify the validity of the thermal model is to assess if there exists dynamic effects, i.e. have the authors seen any dependence on the measured resistance when the current ramps are modified?. Thermal diffusivity towards the substrate and through the current contacts could be modified and so the measured total resistance. I guess that these dynamic measurements are not compatible with MO Kerr effect mapping of FM regions.”

We thank the Reviewer for an interesting suggestion to probe the electro-thermal dynamics associated with the volatile resistive switching in LSMO. Such electro-thermal dynamics could be extremely fast. For example, the insulator-to-metal switching via Joule heating can be induced on sub-nanosecond timescales as demonstrated in *Nat. Nanotechnol.* **9**, 453 (2014). Unfortunately, our MOKE setup and the fabricated large size LSMO devices are not suitable for high-speed measurements. We agree that exploring the electro-thermal dynamics in LSMO is an intriguing topic, but this is far beyond the scope of the current paper.

In conclusion, we provided extensive new experimental data that supports the electro-thermal origin of the observed volatile resistive switching in our LSMO devices and we expanded the literature review to compare our results to previous studies on nonvolatile switching based on ionic electromigration. We hope that the revised manuscript thoroughly addresses the Reviewer’s concerns. We also hope that the Reviewer can consider the magnetism aspects of our work in making the decision about the importance of our results. The search for new approaches to achieve voltage-controlled magnetism remains one of the central efforts in the magnetism community. Our work presents a novel concept of utilizing resistive switching for on/off local magnetic switching. We strongly believe that the implications of our results to the field of magnetism have to be taken into account to make the proper evaluation of the novelty of our work.

Reviewer 2

“The manuscript by Salev et al. reports on resistive switching phenomena in LSMO-based devices. The authors discuss the switching mechanism supported by magneto-optical microscopy. The manuscript is clearly written and structured and the topic is suitable for the journal. However, the presented results are not on the level expected and required for publication in Nature Communications.”

We thank the Reviewer for the criticism as it highlights the important points that require better, more detailed explanations. The criticism, however, is based on misinterpretation of our results using the framework of voltage-driven ionic migration that causes **nonvolatile resistive switching**. The ionic migration in complex oxides has been extensively studied in the past, as the Reviewer points out. Our work describes a fundamentally different type of switching – electrical triggering of a metal-insulator transition that produces **volatile resistive switching**. In the revised manuscript, we clearly separated the volatile and nonvolatile switching types, expanded the discussion on the fundamental novelty of our work, and included new experimental data on room temperature switching in LSMO devices to highlight the possibility of practical applications. We detail those revisions in the responses below. We hope that the new manuscript would clarify the misunderstanding and enable an unhindered communication of our results.

Specific questions raised by Reviewer 2:

1. I see neither some contribution to the fundamental knowledge on resistive switching phenomena, nor on the application side. My general impression is that the authors are not entirely aware on the basic requirements and latest achievement in the field of resistive switching memories.”

Perhaps the original manuscript did not explain clearly the novelty of our work. We hope the new version and the extensive new set of experiments presents this much better. We show that the volatile metal-to-insulator switching occurs in characteristic spatial pattern: the formation of an insulating barrier perpendicular to the current flow. The barrier formation in this type of switching systems has not been observed before as all previous works on the electrical triggering of MIT focused on the percolating filamentary switching. We further argue that the transverse barrier formation is a general feature of low- to high-resistance switching and the barrier formation could occur in nonvolatile switching based on ionic electromigration. We highlight the above points throughout the manuscript:

Page 2: “*It remains unknown, however, whether the $M \rightarrow I$ switching follows any characteristic spatial pattern. In this work, we show that $M \rightarrow I$ switching occurs by the formation of an insulating barrier perpendicular to the current flow, in contrast to the metallic filamentary percolation along the current. We observed the barrier formation experimentally and correlated its appearance with the development of an unusual N-type negative differential resistance (NDR) nonlinearity in the I-V characteristics of a device. Using theoretical analysis, we present evidence that this transverse barrier formation is a universal property of $M \rightarrow I$ switching, making our findings broadly relevant to a whole class of such resistive switching*

systems. Finally, we discuss the implications of our finding to nonvolatile ionic-migration-based low- to high-resistance switching”.

Page 6: “In our model, the only special ingredient enabling the barrier formation is a thermal transition from a low- to a high-resistance state. Because in the simulations we did not have to make any explicit assumption about the phase separation, magnetic properties, defect density profiles, etc., we conclude that our analysis provides a universal description of the voltage-triggered metal-to-insulator phase transition mediated by Joule heating. Many other materials in the manganite family and some magnetic semiconductors could have similar resistive switching behavior”.

Page 7: “Our resistor network simulations show that the insulating barrier that forms during the $M \rightarrow I$ volatile switching concentrates not only the dissipated power but also the applied voltage. It is possible that the transverse barrier formation can occur in systems which feature a nonvolatile low-to-high resistance switching caused by ion electromigration, such as rare-earth nickelates [69,70], manganites [69–74], and cuprates [69,71,75]. In fact, spatial patterns previously reported for the nonvolatile switching in $YBa_2Cu_3O_7$ [75], LSMO [76] and in $NdNiO_3$ [77] can be rationalized by considering the tendency of low- to high-resistance switching to occur by the formation of a transverse insulating barrier”.

2. “There are no resistive switching characteristics that can verify a reliable device operation. The very small hysteresis in Fig. 1b (only for 3 low temperatures) and in Fig. 1c (only for for 60 K) is indicating instable resistive states. At 340 K the relation is linear! The presented I-V and V-I curves are only in the positive range, and only one cycles per temperature is presented. I cannot accept these results as a verification for reliable and stable resistive switching. The currents are high (mA range) and the voltage very high (up to 60 V). Basic resistive switching characteristics are missing: endurance, retention, OFF/ON ratio switching time, evtl. multilevel switching, reliable statistics etc..“

Our work focuses on **volatile** resistive switching, which is different from nonvolatile switching. The volatile switching is induced by holding current or voltage on an MIT material and the switching resets automatically when the stimulus is turned off. What the Reviewer refers to as “instable” is in fact the special property of volatile switching. The fact that the switching in our LSMO devices is volatile was highlighted in the manuscript:

Page 3: “The switching is volatile, i.e. the device automatically resets into the initial low-resistance state when the current is ramped down”.

“...very small hysteresis in Fig. 1b ...”

Small I-V hysteresis is a common feature in volatile switching. The presence or absence of hysteresis, however, is not critical for the volatile switching, in contrast to the nonvolatile switching where the I-V hysteresis is the effect-defining feature.

“...At 340 K the relation is linear...”

The volatile switching in our LSMO devices is due to the electrical triggering of MIT, thus the switching behavior disappears above T_c as the device is already in the high-temperature

insulating phase. This was highlighted in the manuscript:

Page 3: “*The switching is observed over a wide temperature range up to $T_c \approx 340$ K where the strong nonlinearities disappear, which highlights the relation between the switching and the MIT*”.

“...The presented I-V and V-I curves are only in the positive range...”

We included full I-V curves in SI 2 (Fig. S2.4). As expected for volatile switching mediated by Joule heating, the I-V curves are perfectly symmetric with respect to voltage/current polarity change.

“...only one cycles per temperature is presented...”

We included cycling test in SI 2 (Fig. S2.5). We did not observe any noticeable cycle-to-cycle variation. This is expected for volatile switching. After each cycle, the device automatically resets into the initial state, therefore consecutive switching cycles perfectly follow each other.

“...The currents are high (mA range) and the voltage very high (up to 60 V)...”

The switching currents/voltages may appear high due to the large size devices that were made for the purpose of imaging. However, the electric fields and current densities are reasonable, ~ 1 -6 kV/cm and ~ 0.5 -2 MA/cm², respectively.

For comparison, we can consider nonvolatile resistive switching due to oxygen migration in SrFeO₃, which is an antiferromagnetic perovskite oxide (LSMO is a ferromagnetic perovskite oxide) as described in *Adv. Mater.* **31**, 1903391 (2019). The field required to induce the nonvolatile switching in SFO was several thousand kV/cm (several volts over 20-nm-thick vertical devices). The nominal current density through a 20×20 μm² device was up to several kA/cm². However, since a substantial portion of the current is expected to flow through a ~ 1 -μm-wide filament, the current density in the active switching region can be on the order of 0.1-1 MA/cm². Thus, volatile switching in LSMO due to Joule-heating-triggered MIT requires much smaller electric field and comparable current density as nonvolatile resistive switching due to oxygen migration in related complex oxide materials.

We added to the manuscript:

Page 7: “*The switching in our devices requires relatively large driving voltages/currents resulting in large dissipated power, especially at cryogenic temperatures, due to the large device sizes (50×100 μm²) fabricated for MOKE imaging. Reducing the device size down to nanoscale has been shown to dramatically decrease the switching energy down to a few picojoules for MIT-based volatile resistive switching mediated by Joule heating [32,35]*”.

“...endurance...”

We added the demonstration of high endurance of LSMO devices by performing multiple consecutive switching cycles and by subjecting the devices to a long dc voltage stress test. We included new data in SI 2 (Figs. S2.5 and S2.6). High endurance of volatile switching in LSMO is expected. The switching is due to Joule heating bringing the device above $T_c \sim 340$ K. Such temperature is far below the decomposition temperature and the switching can be performed

over a large number of cycles without device degradation.

“...retention...”

Because the resistive switching in LSMO is **volatile**, retention characterization is not applicable. We note that high-speed dynamical effects and even subtle, unconventional memory effects (for example, *Nature* **569**, 388 (2019)) maybe present in volatile switching. However, the studies of ultrafast dynamics of volatile resistive switching in LSMO are far beyond the scope of our work.

“...OFF/ON ratio...”

The on/off resistance switching ratio of the volatile switching in LSMO is fully determined by the resistance temperature dependence, which is a material property (similar to other materials displaying volatile MIT-based switching). The switching occurs from the resistance corresponding to the normal material value at the measurement base temperature into the resistance corresponding to the high-temperature insulating state (as was discussed in the original manuscript). For the reader’s convenience, we also specified the switching ratio at room temperature in the revised manuscript:

Page 7: “*From practical point of view, robust volatile resistive switching with the ratio of $\Delta R/R \sim 300\%$ can be induced in LSMO at room temperature, which is beneficial for practical applications (multiple examples of room temperature characterization can be found in SI 2)*”.

“...switching time...”

Electro-thermal volatile resistive switching due to the MIT triggering can be induced on extremely fast timescales. For example, it has been demonstrated that Joule heating can trigger the insulator-to-metal switching in V_2O_3 on the sub-nanosecond timescale (*Nat. Nanotechnol.* **9**, 453 (2014)). Our large size devices made for the imaging purposes are not compatible with high-speed measurements. Therefore, the switching time measurements in our LSMO devices would show the limitations of electrical circuits and cryostat stability, rather than the intrinsic material response time. Studying the electro-thermal dynamics of volatile resistive switching in LSMO is an interesting topic, but is far beyond the scope of the present work.

“...evtl. multilevel switching...”

Because the resistive switching in LSMO is **volatile** and there is no long-term memory, multilevel switching characterization is not applicable.

“...reliable statistics...”

As can be seen from the cycling test in Fig. S2.5 in SI 2, there is no noticeable cycle-to-cycle variation in volatile resistive switching in LSMO. This behavior is fully expected. Because the switching is volatile, the device resets into the original state and every switching cycle perfectly repeats each other. There is no meaningful switching statistics that could be obtained for the switching in one specific LSMO device. There is of course a device-to-device variability due to fabrication imperfection. In our case, this variability is rather small because of the large device sizes, but it can be expected that reducing the device size down to nanoscale would have a significant impact on the device-to-device variability. Although this is an interesting topic,

finding optimal conditions for reproducible device fabrication is far beyond the scope of the present work.

3. “The fact that the devices are initially low resistive and are further reset to off state is not very unusual. The provided MOKE experiments are clear, but provide no new insight in the RS process. Similar visualization has been provided for e.g. Fe:STO (MRS Proceedings, 1691, Mrss14-1691-bb03-09. doi:10.1557/opl.2014.562). Thus, I cannot acknowledge some new detail on the mechanism revealed.”

The paper by V. Havel *et. al.* describes the **electroforming** in Fe:SrTiO₃, a material that features a **nonvolatile** high- to low-resistance switching due to oxygen migration. Our work focuses on **volatile** switching from a metal to an insulator in LSMO. This volatile switching does not require electroforming, does not involve oxygen migration, and occurs in the opposite direction, from a low-resistance state to a high-resistance state. The electroforming imaging performed by V. Havel *et. al.* does not challenge the novelty of our work. We report the imaging of the characteristic spatial pattern (transverse insulating barrier) that emerges in voltage-triggered electronic phase transition from a metal to an insulator, which has not been done before. By the suggestion of both Reviewers, we included an overview of nonvolatile switching broadening the scope of our paper. Electroforming, on the other hand, is a very specific feature of ionic migration based resistive switching systems. In our opinion, the manuscript would not benefit from a review of the electroforming process.

4. “The authors should be aware that VO_x, PCMO and NbO₂ are only a very small part of RS materials and systems used. Fundamentally important materials such as STO, TaO_x, HfO_x, etc. are even not mentioned.”

Materials, such as VO₂, V₂O₃, V₃O₅, NbO₂, PCMO, feature a volatile switching due to electrical triggering of an insulator-to-metal phase transition, which is related to the triggering of the metal-to-insulator transition in LSMO. Materials such as TiO₂, Ta₂O₅, HfO₂, SrTiO₃, display a nonvolatile switching due to ionic electromigration, which is very different from the volatile switching in LSMO. We highlight this in the revised manuscript:

Page 1: “Many previous studies explored various aspects of nonvolatile switching based on ionic electromigration, which is promising for memory applications [16]. Recently, there has been a great interest in a different type of resistive switching – volatile switching due to electrical triggering of a metal-insulator transition (MIT). Such switching is induced by applying and holding an electrical stimulus to an MIT material and the switching automatically resets back into the initial state upon turning off the stimulus (hence the name volatile)”

To relate our results on the volatile metal-to-insulator switching in LSMO to the previous works on nonvolatile switching, we added a new discussion paragraph:

Page 7: “Volatile and nonvolatile resistive switching types bare certain similarities. The formation of percolating conducting filaments occurs both in I→M volatile switching and in nonvolatile high-to-low resistance switching. The later can be due to oxygen electromigration in binary [47–55] and complex oxides [56,57] or due to metal cation diffusion in conducting

bridge memories based on oxide [58–66] and non-oxide materials [67,68]. Our resistor network simulations show that the insulating barrier that forms during the $M \rightarrow I$ volatile switching concentrates not only the dissipated power but also the applied voltage. It is possible that the transverse barrier formation can occur in systems which feature a nonvolatile low-to-high resistance switching caused by ion electromigration, such as rare-earth nickelates [69,70], manganites [69–74], and cuprates [69,71,75]. In fact, spatial patterns previously reported for the nonvolatile switching in $YBa_2Cu_3O_7$ [75], LSMO [76] and in $NdNiO_3$ [77] can be rationalized by considering the tendency of the low- to high-resistance switching to occur by the formation of a transverse insulating barrier”.

5. “On page 3 is written: “...We found that the metal-to-insulator resistive switching occurs via nucleation and growth of an insulating barrier that spans...” Nucleation (phase formation) and growth are two completely different processes. I also do not see any experimental evidences that either nucleation, or growth are rate limiting.”

We agree with the Reviewer that using terms “nucleation” and “growth” with regards to the formation of a transverse insulating barrier could potentially cause confusion as these terms have well-defined meaning in thermodynamic theory of phase transitions. This is especially true because the volatile switching presented in our work is due to the electrical triggering of a metal-insulator phase transition. To keep language precise, we refrain from using terms “nucleation” and “growth” in the revised manuscript. Instead, we use “formation” and “expansion” with regards to the barrier appearance and increase of its size with increasing voltage.

6. “The authors are not discussing at all the possibility of oxygen exchange that is supposed to occur at these high voltages and currents. Is the possible loss/enrichment of oxygen also relevant to the resistance change?”

We further analyzed the existing data and presented new data in SI2. We did not find any evidence that oxygen migration plays any significant role in the volatile resistive switching in our LSMO devices. We direct the reader to SI 2 in the main text:

Page 3: “The detailed discussion about the origin of volatile resistive switching in LSMO is available in SI 2. We compared the resistance values in I - V curves to resistance-temperature (R - T) dependence, tested the influence of the switching on the R - T dependence, analyzed the switching power vs. temperature and the switching voltage vs. temperature dependencies, tested for bipolar switching, performed cycling and voltage stress tests, and probed the influence of oxygen partial pressure on the switching behavior. All our results point to the single conclusion that the observed volatile switching in LSMO originates from the MIT triggering mediated by Joule heating, while no indication of electrically driven oxygen migration has been found”.

7. “The I - V plot in Fig. 3 is not presenting any resistive switching. It is an ordinary current-voltage dependence with no indication on resistive switching.”

The I - V plots in Fig. 3 (similar to Fig. 1 b and c and Fig. 2b) show **volatile resistive switching** caused by electrical triggering of a metal-to-insulator phase transition. The current-controlled

I-V (red curve) displays abrupt switching/jump that is due to the switching in the entire LSMO device. The voltage-controlled I-V (blue curve) shows a gradual evolution with the development of the N-type NDR region. This NDR is due to an abrupt metal-to-insulator switching that is localized within a transverse insulating barrier. We highlight this point in the manuscript:

Page 3: *“The appearance of voltage-induced insulating barrier proves that the LSMO device undergoes an abrupt resistive switching on a microscopic level, from the metallic into the insulating phase, even though the global I-V curve of the entire device displays smooth, gradual, low- to high-resistance evolution”.*

8. *“Most experiments were performed at low temperatures that are not advantageous for applications.”*

The transport measurements in our work were performed in a broad temperature range, 60-340 K, which includes room temperature. The cryogenic temperatures in the MOKE measurements (100 K and 250 K) were chosen to enhance the imaging contrast. We included new data further demonstrating the device operation at room temperature in SI 2. We also briefly discuss room temperature operation, advantages of downscaling, and possible applications for volatile switching in the main text:

Page 1: *“MIT-based switching is often accompanied by a large change of electrical transport and optical properties making it attractive for applications in rf electronics [17,18], optoelectronics [19–21], and biologically inspired artificial neurons [10–13]”.*

Page 7: *“From practical point of view, robust volatile resistive switching with the ratio of $\Delta R/R \sim 300\%$ can be induced in LSMO at room temperature, which is beneficial for practical applications (multiple examples of room temperature characterization can be found in SI 2). The switching in our devices requires relatively large driving voltages/currents resulting in large dissipated power, especially at cryogenic temperatures, due to the large device sizes ($50 \times 100 \mu\text{m}^2$) fabricated for MOKE imaging. Reducing the device size down to nanoscale has been shown to dramatically decrease the switching energy down to a few picojoules for MIT-based volatile resistive switching mediated by Joule heating [32,35]”.*

We hope that the extensive revisions of the manuscript and SI as well as the detailed answers in this reply thoroughly address the Reviewer’s concerns. We ask the Reviewer to consider the differences and the similarities between volatile and nonvolatile resistive switching types to, on one hand, accurately evaluate the switching properties of LSMO, and on the other hand, to recognize the general implications of establishing the transverse barrier formation as the characteristic property of low- to high-resistance switching. We also hope that the Reviewer can consider the magnetism aspects of our work in making the decision about the importance of our results. The search for new approaches to achieve voltage-controlled magnetism remains one of the central efforts in the magnetism community. Our work presents a novel concept of utilizing resistive switching for on/off local magnetic switching. We strongly believe that the implications of our results to the field of magnetism have to be taken into account to make the proper evaluation of the novelty of our work.

Reviewers' Comments:

Reviewer #1:

Remarks to the Author:

The authors have strongly modified the previous manuscript by considering all the criticisms included in my previous report and they have prepared a very detailed answer to all the questions and requests made by the two referees, including new experimental results which help to clarify the many raised questions and the title of the manuscript.

The content of the present manuscript reports convincing new results linking the volatile resistive switching (RS) associated to a MIT with the magnetic switching visualized by MOKE. The manuscript now is much more complete in the presentation of the state of the art about the RS phenomena and the different mechanisms associated to non-volatile RS and the volatile RS presented here as a novel result. The discussion about the electro-thermal origin of this novel RS appears convincing and relevant.

I consider that in the present state the manuscript is worth of being published in Nature Communications.

Minor corrections:

- Figure S3 in S.I. still includes the term "domain nucleation" which within the text has been corrected as "domain formation".

Reviewer #2:

Remarks to the Author:

I carefully read the rebuttal and the new data provided by the authors and appreciate their time and efforts. However, the provided clarifications and new data are not convincingly supporting the main (Joule heating induced switching) thesis. Other major issues also remain open. The provided responses strengthens my impression that the authors are not familiar with the field of resistive switching memories. I cannot recommend the manuscript for publication.

General Comments

A. Please, read the literature on volatile resistive switching (for example Adv Mater. 29 (2017) 1604457, as well many others).

B. The authors stated that their on/off ratio is "beneficial for practical applications". This is definitely not true. The off/on ratio is not sufficient to claim even average performance. (See comment A)

C. The linear I-V relation cannot be considered as a switching (irrespective volatile or nonvolatile)! The linear I-V relation is typical for materials that do not show resistive switching e.g. metals.

D. There is no unequivocal experimental proof that the RS effect is solely based on Joule heating. At these currents, Joule heating is surely playing a role, but it also allows ions to migrate faster... The arguments that aim to eliminate oxygen ion (vacancy) switching mechanism are not convincing. Some examples from the rebuttal. The authors provided arguments claiming they support solely Joule-mediated switching. To argument #1 – RS memories that are forming-free based on ion-migration (both ECM and VCM) are often reported... To argument #2 – same applies to ion-migration based volatile switching... To argument #3 – same is applying to ion-migration based volatile switching... etc.. Please refer also to comment A.

E. The authors are using symmetric electrodes. It is therefore no wonder that the I-V curves are symmetric. This is not a proof for Joule-heating determined switching. See comment A.

F. The authors are arguing that forming the barrier in direction perpendicular to the current is unique and provides new contribution to the fundamental knowledge. This is not correct. Using metals of high oxygen affinity always leads to formation of oxide film (often more insulating than the switching film) and the direction is always perpendicular to the film. This is a usual phenomenon by electrochemical oxidation of metals.

Comments

1. Endurance – the author presented 100 cycles, claiming high endurance. This is by orders of magnitude not sufficient to claim good endurance. See comment A.
2. The authors stated their current densities are within the typical values. This is not entirely correct. Vertical devices (LSMO) with 20 nm- 50 nm film thickness will surely show much higher current densities. At least this is the experience with other RS materials. The fields are indeed within the range, however, appears the question why at these fields no ion migration occurs? (see also comment D)
3. On the statistical deviation – the 100 cycles shown and the way of presentation is not verifying in any sense good statistics. Please, refer to comment A.
4. The authors stated that oxides such as HfO₂, Ta₂O₅ etc. are not used for non-volatile switching. This is not true. Please, refer to comment A.
5. The point that many materials (devices) are showing initial on state and then transit to off state is still valid. No new (or rare) phenomena is reported.
6. As before in Fig. 3 there is no clear indication on RS.
7. High (room temperature) measurements are not indicating resistive switching. Linear (or nearly linear) I-V dependences are cannot be considered as RS.

Reply to Reviewers

Manuscript: Transverse barrier formation by electrical triggering of a metal-to-insulator transition

NCOMMS-20-41785A-Z

Reviewer 1

“The authors have strongly modified the previous manuscript by considering all the criticisms included in my previous report and they have prepared a very detailed answer to all the questions and requests made by the two referees, including new experimental results which help to clarify the many raised questions and the title of the manuscript.

The content of the present manuscript reports convincing new results linking the volatile resistive switching (RS) associated to a MIT with the magnetic switching visualized by MOKE. The manuscript now is much more complete in the presentation of the state of the art about the RS phenomena and the different mechanisms associated to non-volatile RS and the volatile RS presented here as a novel result. The discussion about the electro-thermal origin of this novel RS appears convincing and relevant.

I consider that in the present state the manuscript is worth of being published in *Nature Communications*.

Minor corrections:

- Figure S3 in S.I. still includes the term “domain nucleation” which within the text has been corrected as “domain formation”.

We thank the Reviewer for all the comments and suggestions that improved the manuscript. We are glad that the Reviewer finds our work suitable for the publication in *Nature Communications*. We thank the Reviewer for very careful reading of the manuscript and finding inaccurate labels in Fig. S3. We corrected those labels to “barrier formation”.

Reviewer 2

“Please, read the literature on volatile resistive switching (for example Adv Mater. 29 (2017) 1604457, as well many others).”

“The authors stated that oxides such as HfO₂, Ta₂O₅ etc. are not used for non-volatile switching. This is not true. Please, refer to comment A.”

The paper pointed out by the Reviewer describes a threshold switching in a CBRAM-based device which:

1. Occurs from a high to low resistance.
2. Originates from the metal cation migration.
3. Results in the formation of a longitudinal conducting filament.

Our work presents the volatile switching that:

1. Occurs from a low to high resistance.
2. Originates from the triggering of metal-insulator transition.
3. Results in the formation of a transverse insulating barrier.

We describe a completely different switching type. The previously reported threshold switching in CBRAM-based devices does not challenge the novelty of our work.

In the revised manuscript, we further highlight the different origin of the switching in LSMO as compared to the threshold switching in CBRAM-based devices and provide a reference to a review paper on the threshold switching (ref. 46) to guide the interested reader. We added:

Page 3: “*All our results point to the single conclusion that the observed volatile switching in LSMO originates from the MIT triggering mediated by Joule heating, which is a common switching mechanism in MIT-based devices [28,44,45]. We found no evidence that ionic migration, which under special conditions could produce volatile threshold switching [46], play any significant role in our experiments*”.

“The authors stated that their on/off ratio is “beneficial for practical applications”. This is definitely not true. The off/on ratio is not sufficient to claim even average performance. (See comment A)”

While it is true that many resistive switching systems have extremely high ON/OFF ratio, it is not obvious that a high ON/OFF ratio is needed in every possible application. For example, GMR- and TMR-based devices with the ON/OFF ratio of ~10% and ~100%, respectively, are actively used in industry. Fully functioning hardware-level neural networks has been demonstrated using TiO₂ memristors with ~900% switching range, Nature **521**, 61 (2015) and Nat. Comm. **9**, 2331 (2018). In our work, the ON/OFF ratio of ~300% can be achieved in LSMO devices at room temperature, which allows easy and reliable detection using even the most basic circuitry. Furthermore, the volatile resistive switching in LSMO drives the magnetic switching (which is a unique functionality), has high endurance and no cycle-to-cycle variability. All of the above properties could be beneficial for practical applications. We added to the manuscript:

Page 7: “*From practical point of view, volatile switching in LSMO has multiple beneficial*

properties. Resistive switching with the ratio of $\Delta R/R \sim 300\%$ and coinciding magnetic switching can be induced at room temperature (multiple examples of room temperature characterization are available in SI 2). The switching shows no noticeable cycle-to-cycle variability (Figs. S2.5 and S2.7), which is a common problem in resistive switching devices [50]. LSMO devices also have excellent endurance: no sign of device degradation has been observed over the 8-hour-long dc voltage stress test (Fig. S2.6) and over 5×10^6 fast switching cycles (Fig. S2.7)".

“The linear I-V relation cannot be considered as a switching (irrespective volatile or nonvolatile)! The linear I-V relation is typical for materials that do not show resistive switching e.g. metals.”

“As before in Fig. 3 there is no clear indication on RS.”

“High (room temperature) measurements are not indicating resistive switching. Linear (or nearly linear) I-V dependences are cannot be considered as RS.”

The I-V curves in LSMO devices are not linear and clearly exhibit resistive switching:

The above curves were recorded at room temperature (taken from Fig. S2.4). The I-V curves in Fig. 3 were recorded at 250 K and, therefore, exhibit more pronounced nonlinearities and a higher ratio switching. We do not understand why the Reviewer considers such I-V curves to be linear (i.e. showing the straight line dependence) or how such I-V curves can be measured in normal metals.

“There is no unequivocal experimental proof that the RS effect is solely based on Joule heating. At these currents, Joule heating is surely playing a role, but it also allows ions to migrate faster...”

The arguments that aim to eliminate oxygen ion (vacancy) switching mechanism are not convincing. Some examples from the rebuttal. The authors provided arguments claiming they support solely Joule-mediated switching. To argument #1 – RS memories that are forming-free based on ion-migration (both ECM and VCM) are often reported... To argument #2 – same applies

to ion-migration based volatile switching... To argument #3 – same is applying to ion-migration based volatile switching... etc.. Please refer also to comment A.”

“The authors are using symmetric electrodes. It is therefore no wonder that the I-V curves are symmetric. This is not a proof for Joule-heating determined switching. See comment A.”

Ion migration origin of the switching phenomenon we observed in LSMO is unrealistic. If we assume that the ion migration drives the switching then:

1. When the voltage is applied, the entire $50 \times 100 \mu\text{m}^2$ device becomes depleted of oxygen or gets infused with Pd from the electrodes (MOKE measurements show that the entire device could be switched).
2. When the voltage is turned off, the entire $50 \times 100 \mu\text{m}^2$ device spontaneously “re-assembles” back into the perfect perovskite lattice, so neither MIT nor ferromagnetism are affected by performing the switching cycles (the two properties that are extremely sensitive to stoichiometry, defects and disorder).

We highlighted this point by adding a new paragraph in SI 2:

Page 4 (SI): *“MIT and ferromagnetism in LSMO are extremely sensitive to stoichiometry, defects and disorder [2]. However, neither MIT nor ferromagnetism are affected by cycling the LSMO device back and forth through the volatile resistive switching, even when the switching is induced in the entire $50 \times 100 \mu\text{m}^2$ device under the application of large voltages/currents (see Figs. 2 and 3 in the main text and Fig. S3). If the ionic migration was responsible for the observed switching, it would be impossible that the stoichiometry and crystal structure could spontaneously restore to the pristine state within the $50 \times 100 \mu\text{m}^2$ device area when the voltage/current is turned off and no indication of chemical/structural change during the switching could be found in the MIT and ferromagnetic properties. The observed behavior, on the other hand, is consistent with the Joule heating origin of the switching. Warming up the device just past $T_c \approx 340 \text{ K}$ is not expected to cause material decomposition, thus neither MIT nor ferromagnetic properties are affected by the switching”.*

We provide a comprehensive list in SI 2 of 14 experimental observations that are consistent with the electro-thermal origin of the volatile switching in LSMO. As the Reviewer points out, some of those properties could sometimes be found in systems that exhibit electro-chemical switching. However, there are no electro-chemical switching systems that exhibit all of the observed properties at the same time:

1. No electroforming.
2. Volatile switching.
3. No impact on the R-T.
4. The switching disappears above the T_c of the thermal MIT.
5. The switching resistance values perfectly coincides with the values in the thermal MIT.
6. The switching is accompanied by the coupled magnetic transition.
7. Neither the MIT nor ferromagnetism are affected by performing switching cycles.
8. The switching power increases monotonically with decreasing temperature. The switching field has a non-monotonic temperature dependence.

9. Perfectly symmetric I-V curves.
10. Perfectly reproducible I-V curves over an 8-hour-long quasi-dc cycling.
11. No resistance change/drift over an 8-hour-long high dc voltage stress test.
12. Perfectly reproducible I-V curves over 5×10^6 fast switching cycles.
13. Switching unaffected by oxygen partial pressure.
14. Qualitatively different switching type (different I-Vs, large cycle-to-cycle variability, strong impact on the MIT, etc.) can be induced after the application of a too high voltage, i.e. after performing electroforming to initiate ionic migration.

“The authors are arguing that forming the barrier in direction perpendicular to the current is unique and provides new contribution to the fundamental knowledge. This is not correct. Using metals of high oxygen affinity always leads to formation of oxide film (often more insulting than the switching film) and the direction is always perpendicular to the film. This is a usual phenomenon by electrochemical oxidation of metals.”

“The point that many materials (devices) are showing initial on state and then transit to off state is still valid. No new (or rare) phenomena is reported.”

Our work presents a different switching mechanism from redox reactions at the oxide/electrode interface. The switching in LSMO does not occur at the electrode, the transverse barrier forms deep inside the LSMO device. This is completely different phenomenology, which indicates different switching origin in LSMO.

“Endurance – the author presented 100 cycles, claiming high endurance. This is by orders of magnitude not sufficient to claim good endurance. See comment A.”

“On the statistical deviation – the 100 cycles shown and the way of presentation is not verifying in any sense good statistics. Please, refer to comment A.”

In the revised manuscript, we added the demonstration of 5×10^6 switching cycles (Fig. S2.7). We observed perfectly repeatable I-V curves having ~1% parameter variations, which is within the measurement accuracy. We did not observe any sign of device degradation. We added new Fig. S2.7 and a new discussion paragraph in SI 2:

Page 6 (SI): “*The switching in LSMO shows high endurance and no cycle-to-cycle variability over 5×10^6 high-speed switching cycles. The electrical circuit used in the high-speed measurements is shown in Fig. 2.7a. We employed the combination of a function generator and an amplifier in order to create a triangular-shape 15-ms-period voltage waveform that has a large enough amplitude to induce the full switching in a $50 \times 100 \mu\text{m}^2$ device. Measured dynamic I-V curves are shown in Fig. 2.7b. The dynamic I-V curves have similar appearance to the dc I-V curves (for example in Fig. S2.4a). The larger hysteresis in the dynamic I-V curves is consistent with the Joule heating origin of the switching as thermal equilibrium in a large-size device cannot be established quickly. All the I-V curves recorded between the 1st and 5×10^6 -th cycles are identical resulting in no apparent dependence of the low- and high-resistance states on cycling (Fig. S2.7c). The*

statistical analysis of the low- and high-resistance states over the switching cycles gives $R_{low} = 1150 \pm 16 \Omega$ and $R_{high} = 4050 \pm 10 \Omega$. The observed resistance cycle-to-cycle deviations (16Ω and 10Ω) are well within the accuracy of the “single-shot” oscilloscope measurements, indicating the absence of the device degradation and extremely high repeatability of the volatile resistive switching in LSMO. While high-endurance switching can be achieved in systems based on ionic migration, the lack of variability in the I-V shape or the low- and high-resistance states over 5×10^6 cycles suggests that the observed volatile switching in LSMO is not related to ionic migration”.

“The authors stated their current densities are within the typical values. This is not entirely correct. Vertical devices (LSMO) with 20 nm- 50 nm film thickness will surely show much higher current densities. At least this is the experience with other RS materials. The fields are indeed within the range, however, appears the question why at these fields no ion migration occurs? (see also comment D)”

We report experimental observations:

1. The current densities in our LSMO devices are comparable to the typical values found in ion-migration-based devices ($\sim 1 \text{ MA/cm}^2$).
2. The electric fields are much smaller than in ion-migration based devices ($\sim 1 \text{ kV/cm}$ vs. $\sim 1 \text{ MV/cm}$).

The volatile switching in our LSMO device can be induced throughout the entire device. Therefore, there is no reason to speculate or expect a radically different behavior in hypothetical devices of different dimensions.

Reviewers' Comments:

Reviewer #2:

Remarks to the Author:

The authors have further extended their responses, made additional experiments and addressed previous comments as well corrected some imprecise statements. I now understand part of the confusions and my criticism were caused by the term metal to insulator transition (MIT) used as a synonym solely for Mott-transitions. I understand that MIT is used in some groups as a synonym for Mott-metal to insulator transitions, but this leads to confusions. Many other RRAM devices show MIT phenomenon, despite having other origin. Therefore, I strongly suggest to clearly explain from the very beginning that this manuscript deals only with Mott-type MIT.

I have read the rebuttal and the manuscript again and still cannot recommend it for publication. The manuscript reports interesting phenomenon, but in my view (see detailed comments below) the fundamental insight and especially the prospective for possible applications are not strong enough to justify publication in Nature Communications.

Addressed comments:

1. I fully agree with the authors that 3-times difference in the resistance OFF/ON ratio is absolutely sufficient, provided the low variability.
2. I agree with the authors that the behaviour of the devices strongly indicates that the switching is not caused by ion migration. Lower electric fields and same characteristics in air and in vacuum are also supporting this conclusion. I would rather prefer having some structural changes registered by e.g. in situ X-ray absorption analysis, or other technique that directly proves the Mott-transition, but I accept the argumentation by the authors.
3. I agree that the type of transition is unusual for Mott-MIT.

Comments:

4. The devices cited in my previous comments are indeed ECM/CBRAM devices, which have different mechanism of volatile switching (compared to Mott-type). In fact all type ReRAM devices (filamentary and non-filamentary) show volatile switching. Despite the mechanism is different the (volatile) switching characteristics of different type devices should be compared. Taking into account CBRAM volatile switching characteristics (Adv Mater. 29 (2017)1604457) one can easily acknowledge that these are by orders of magnitude better compared to the ones shown in the present manuscript.

5. The device characteristics are also not good in terms of switching voltages (up to 75 V) and currents (tens of mA). This also predetermines high power consumption.

6. 5×10^6 cycles is a good endurance but still by orders of magnitude lower compared to e.g. CBRAM volatile devices. The switching speed is also lower (ms range), whereas other e.g. CBRAM devices show ns-range responses.

7. About the linearity of the switching dependences. In fig 1. b and 1 c are shown resistive switching dependences. At higher temperatures the lines in dark red, red and orange colours do not show switching. The dependence marked in dark red colour is clearly linear! Only at low temperatures dependences, marked in green, light blue and blue switching is clear. Please comment.

8. The authors should take into account that lateral devices of the type they present in this manuscript are difficult to be used in applications. Devices need to be downscaled and typically vertical arrangement is used. This makes the conditions quite different compared to the large lateral cells used in the study. Therefore, one should estimate how a "hypothetical" device will behave. It should be ensured that changing the configuration (from lateral to vertical) and size will

not change the conditions and type of switching. Please, note that several groups have studied LSMO based vertical devices and found only VCM-type switching.

Reply to Reviewers

Manuscript: Transverse barrier formation by electrical triggering of a metal-to-insulator transition

NCOMMS-20-41785B

Notations

Reviewer's comments are in blue.

Authors' response is in black.

Manuscript and SI quotes are in italic.

Reviewer 2

“The authors have further extended their responses, made additional experiments and addressed previous comments as well corrected some imprecise statements. I now understand part of the confusions and my criticism were caused by the term metal to insulator transition (MIT) used as a synonym solely for Mott-transitions. I understand that MIT is used in some groups as a synonym for Mott-metal to insulator transitions, but this leads to confusions. Many other RRAM devices show MIT phenomenon, despite having other origin. Therefore, I strongly suggest to clearly explain from the very beginning that this manuscript deals only with Mott-type MIT.”

We thank the Reviewer for continuing to provide feedback on our manuscript and we are happy that the source of misunderstanding about the origin of the volatile switching in LMSO has been identified. As the Reviewer suggested, we added a clarification in the introduction paragraph:

Page 1: “*Recently, there has been a great interest in a different type of resistive switching: volatile switching due to electrical triggering of a metal-insulator transition (MIT), i.e. an intrinsic phase transition that alters the charge transport properties of a material (for example, Mott or Peierls transition).*”

Many materials exhibiting volatile MIT-based switching indeed have a Mott transition, however, the MIT in LSMO is driven by the double exchange mechanism, which couples electrical and magnetic properties. In brief, the electron transfer between Mn^{3+} and Mn^{4+} ions (i.e. conductivity) occurs only when the magnetic moments of the ions are parallel (ferromagnetic metal state). Above T_c , the ferromagnetic ordering is lost and the electron transfer can no longer occur leading to electron localization (paramagnetic insulator state). The manuscript provides references to guide interested readers to the original works describing the double exchange mechanism:

Page 2: “*Under equilibrium conditions (i.e. without application of high voltage/current), the devices have two coupled phase transitions at $T_c \approx 340$ K: from a low-temperature ferromagnetic metal to a high-temperature paramagnetic insulator (Fig. 1a). The coupling between the two transitions, magnetic and MIT, is mediated by the double exchange mechanism [42,43].”*

For this work, the intricacies of the phase transition in LSMO (magnetic correlation driven splitting of t_{2g} and of e_g orbitals, Jahn-Teller distortions, polaron formation, etc.) do not impact the result interpretation and are not discussed in the manuscript. The relevant information emphasized throughout the manuscript includes (i) the MIT can be induced thermally, therefore volatile switching by Joule heating is possible, (ii) the MIT has magnetic origin, therefore magnetic imaging can be used to visualize the switching. We further emphasize the second point in the revised manuscript:

Page 4: “*Because of the direct connection between the magnetic and electric properties in LSMO, our measurements imply that the resistive switching from a metal into an insulator does not occur uniformly throughout the device.*”

“I have read the rebuttal and the manuscript again and still cannot recommend it for publication. The manuscript reports interesting phenomenon, but in my view (see detailed comments below) the fundamental insight and especially the prospective for possible applications are not strong enough to justify publication in Nature Communications.”

Our work describes a new type of volatile resistive switching. We agree with the Reviewer that volatile switching has been extensively studied in a variety of materials, including CBRAM devices (e.g. Ag:HfO₂), oxide memristors (e.g. TiO₂) and metal-insulator phase transition materials (e.g. VO₂). Previous studies, however, focused on the volatile switching from a high- to a low-resistance state. Our work investigates the opposite type of volatile switching: from a low- to a high-resistance. While there are multiple materials where a nonvolatile switching can be induced from the initial low-resistance state, the low-to-high resistance volatile switching is highly unusual, as the Reviewer pointed out in the comments. Our work:

- Presents a new phenomenon – volatile metal-to-insulator switching.
- Elucidate the origin of this switching – MIT triggering mediated by Joule heating.
- Investigates the switching mechanism – transverse insulating barrier formation.
- Describes a novel functionality – localized on/off switching of ferromagnetism.
- Demonstrates the potential for applications – room temperature operation, good dc stress and switching endurance, low cycle-to-cycle variability.

The above list constitutes a novel direction in the field of resistive switching, not an incremental advancement on an already well-explored phenomenon. We believe that the results of our work would attract attention in physics, materials science and engineering communities, thus, we seek the publication in *Nature Communications*, a journal with a wide and diverse audience.

“I fully agree with the authors that 3-times difference in the resistance OFF/ON ratio is absolutely

sufficient, provided the low variability.

I agree with the authors that the behaviour of the devices strongly indicates that the switching is not caused by ion migration. Lower electric fields and same characteristics in air and in vacuum are also supporting this conclusion. I would rather prefer having some structural changes registered by e.g. in situ X-ray absorption analysis, or other technique that directly proves the Mott-transition, but I accept the argumentation by the authors.

I agree that the type of transition is unusual for Mott-MIT.”

We thank the Reviewer for agreeing that our measurements identify the triggering of a metal-insulator transition as the origin of the resistive switching in LSMO and that such a switching is an unusual phenomenon.

“The devices cited in my previous comments are indeed ECM/CBRAM devices, which have different mechanism of volatile switching (compared to Mott-type). In fact all type ReRAM devices (filamentary and non-filamentary) show volatile switching. Despite the mechanism is different the (volatile) switching characteristics of different type devices should be compared. Taking into account CBRAM volatile switching characteristics (Adv Mater. 29 (2017)1604457) one can easily acknowledge that these are by orders of magnitude better compared to the ones shown in the present manuscript.

The device characteristics are also not good in terms of switching voltages (up to 75 V) and currents (tens of mA). This also predetermines high power consumption.

5×10^6 cycles is a good endurance but still by orders of magnitude lower compared to e.g. CBRAM volatile devices. The switching speed is also lower (ms range), whereas other e.g. CBRAM devices show ns-range responses.”

Our work focuses on exploring fundamental properties of a new type of resistive switching: volatile switching from a ferromagnetic metal into paramagnetic insulator occurring by the transverse barrier formation. Because of the unique set of features, a direct comparison of the volatile switching in LSMO to the threshold switching in CBRAM devices is not appropriate.

LSMO is metallic in equilibrium (high conductivity) and becomes insulating (low conductivity) when a threshold current is reached. CBRAM threshold switches are insulating and become highly conducting when a threshold voltage is applied. Essentially, the two switching types provide opposite functionalities. If an application requires the transfer of small currents and attenuation of large currents, the switching in LSMO is the solution. If the opposite is needed, i.e. blocking voltages/currents below a certain threshold, then CBRAM threshold switching or an insulator-to-metal volatile switching could be utilized.

The LSMO device geometry for direct MOKE imaging of the barrier formation requires open-surface planar devices of $50 \times 100 \mu\text{m}^2$ size. High switching voltages/currents, high power consumption, slow switching speed, etc. are direct consequences of the device size. Because the volatile MIT-based switching does not originate from the interface/junction properties, the

switching parameters scale with device size. Ultrafast energy-efficient volatile switching has been demonstrated in nanodevices where the applied voltage/current triggers an insulator-to-metal phase transition. It is reasonable to expect that similar scaling behavior of the switching voltage, current, power, time, etc. is also present in the volatile metal-to-insulator switching.

We modified both the main text and SI to highlight the above points:

Page 7: *“From a practical point of view, volatile switching in LSMO has a rare combination of beneficial properties. Our devices show the switching ratio of $\Delta R/R \sim 300\%$ at room temperature. While higher ratios can be achieved in volatile threshold switches, for example in CBRAM-based devices [46], the switching in LSMO occurs from a low to a high resistance state. The low-to-high resistance switching is a unique feature among the volatile switching devices and it could facilitate a new way of utilizing such devices in practical applications. The switching in our devices shows no noticeable cycle-to-cycle variability (Figs. S2.5 and S2.7), which is a common problem in resistive switching devices [50], and has an excellent endurance as no sign of device degradation has been observed over an 8-hour-long dc voltage stress test (Fig. S2.6) and over 5×10^6 fast switching cycles (Fig. S2.7). Moreover, the resistive switching in LSMO drives the switching of ferromagnetism, which constitutes a special functionality that could be relevant to bridging the gap between conventional charge-based electronics and spintronics. Because of the large device sizes fabricated for MOKE imaging ($50 \times 100 \mu\text{m}^2$), the switching in our devices requires large driving voltages/currents resulting in a relatively slow switching speed and large dissipated power, especially at cryogenic temperatures. Reducing the device size down to nanoscale has been shown to enable sub-nanosecond switching times and dramatically reduce the switching energy down to a few picojoules in MIT-based volatile resistive switching mediated by Joule heating [32,35].”*

SI 2, page 6: *“Because of the large size devices optimized for MOKE imaging ($50 \times 100 \mu\text{m}^2$), we had to use a combination of a function generator and an amplifier to produce large enough voltage/current to induce the switching of the entire device. The performance of the high-voltage amplifier determined the limit of how fast a switching cycle can be performed (15-ms-period waveform) and ultimately set the limit of how many switching cycles (5×10^6) can be acquired in a reasonable amount of time. Reducing the device size down to nanoscale dimensions should reduce the switching voltage/current enabling fast speed measurements to determine the ultimate switching time in LSMO and to probe the device endurance past several million cycles.”*

“About the linearity of the switching dependences. In fig 1. b and 1 c are shown resistive switching dependences. At higher temperatures the lines in dark red, red and orange colours do not show switching. The dependence marked in dark red colour is clearly linear! Only at low temperatures dependences, marked in green, light blue and blue switching is clear. Please comment.”

The fact that the switching disappears above the MIT temperature (I-V curves become linear) is one of the main evidence that the switching is due to the triggering of the MIT, as was discussed both in the main text and in SI. When LSMO is already in the insulating state (i.e. at high ambient temperature), the metal-to-insulator switching cannot happen. We thank the Reviewer for pointing

out that the above argument was not clearly outlined in the manuscript. We rephrased the discussion about the temperature dependence of the I-V curves in the main text:

Page 3: *“Distinct switching, i.e. an abrupt “jump” in an I-V curve, can be observed at all temperatures below 310 K. As the temperature approaches the phase transition, in 310-340 K range, the I-V curves are nonlinear but do not display an abrupt discontinuity. Above the transition ($T > T_c \approx 340$ K), the switching completely disappears and the I-V curves become linear indicating the close relation between the switching and the MIT: the metal-to-insulator switching cannot be induced when the LSMO device is already in the insulating state.”*

“The authors should take into account that lateral devices of the type they present in this manuscript are difficult to be used in applications. Devices need to be downscaled and typically vertical arrangement is used. This makes the conditions quite different compared to the large lateral cells used in the study. Therefore, one should estimate how a “hypothetical” device will behave. It should be ensured that changing the configuration (from lateral to vertical) and size will not change the conditions and type of switching. Please, note that several groups have studied LSMO based vertical devices and found only VCM-type switching.”

There is no fundamental reason why the volatile switching in LSMO cannot be induced in nanoscale vertical devices since the switching originates from the intrinsic material property, the metal-insulator phase transition. From the practical point of view, nanofabrication involving complex oxides can be a considerable challenge because of the sensitivity of such materials to defects. However, as long as the fabricated LSMO devices, planar or vertical, exhibit the MIT, it should be possible to induce the volatile switching. As an example, we can consider VO₂, a material that features a volatile resistive switching based on the electrical triggering of an intrinsic insulator-to-metal phase transition (i.e. switching in the opposite direction from LSMO). The majority of studies use planar VO₂ devices as such geometry allows an easy access for various measurement techniques. It is possible, however, to fabricate vertical VO₂ devices in a crossbar geometry and achieve the same set of functional properties as in the planar devices, see for instance J. Appl. Phys. **115**, 154502 (2014). We added to the main text:

Page 7: *“Because MIT-based switching relies on the intrinsic material properties, i.e. triggering of a phase transition, such switching is possible in devices of various geometries including a vertical crossbar [51], a geometry most suitable for dense circuit integration.”*

Reviewers' Comments:

Reviewer #2:

Remarks to the Author:

The authors have addressed my comments. From application point of view, there are serious challenges and I am not convinced that preparing vertical or nanoscale LSMO devices is something that can be reliably achieved. This is a weak point that cannot be resolved without additional experiments and practical demonstration. However, the fundamental aspect is of sufficient importance to be reported in the journal and I recommend publication of the manuscript in its present form.

Reply to Reviewers

Manuscript: Transverse barrier formation by electrical triggering of a metal-to-insulator transition

NCOMMS-20-41785C

Notations

Reviewer's comments are in blue.

Authors' response is in black.

Manuscript and SI quotes are in italic.

Reviewer 2

“The authors have addressed my comments. From application point of view, there are serious challenges and I am not convinced that preparing vertical or nanoscale LSMO devices is something that can be reliably achieved. This is a weak point that cannot be resolved without additional experiments and practical demonstration. However, the fundamental aspect is of sufficient importance to be reported in the journal and I recommend publication of the manuscript in its present form.”

We thank the Reviewer for recommending the publication of the manuscript. We agree that preparing nanoscale vertical LSMO devices could be challenging. Experimental demonstration indeed can provide a definite answer whether it is possible to reliably fabricate crossbar LSMO devices that exhibit volatile metal-to-insulator switching. Previous works, for example Phys. Rev. Mater. **1**, 024401 (2017) and Phys. Rev. B **21**, 214421 (2010), have demonstrated the feasibility of nanoscale patterning and vertical device fabrication without destroying the magnetic and electronic properties of LSMO. When a nanoscale device has a metal-insulator transition, it should be possible to trigger this transition electrically via Joule heating and produce a volatile resistive switching, similar to the results presented in this work.

To highlight the above point, we added to the main text:

Page 6: “*Nanoscale patterning [53] and fabrication of vertical magnetoresistance devices [54] has been demonstrated using LSMO. Thus, successful implementation of nanoscale vertical resistive switching LSMO devices should be possible as well.*”